# A digital twin for real-time biodiversity forecasting with citizen science data

Otso Ovaskainen [1,22] ✉, Steven Winter[2,22], Gleb Tikhonov [3,22], Patrik Lauha [3,22], Ari Lehtiö [4,22], Ossi Nokelainen [1,5], Nerea Abrego [1], Anni Aroluoma[6], Jesse Patrick Harrison [7], Mikko Heikkinen [8], Aleksi Kallio[7], Anniina Koliseva[6], Aleksi Lehikoinen [8], Tomas Roslin [9,10], Panu Somervuo [3], Allan Tainá Souza [11], Jemal Tahir[7], Jussi Talaskivi [4], Alpo Turunen [8], Aurélie Vancraeyenest[7], Gabriela Zuquim[7], Hannu Autto[12], Jari Hänninen [13], Jasmin Inkinen [13], Outa Kalttopää[12], Janne Koskinen[14], Matti Kotakorpi[15], Kim Kuntze [16], John Loehr[15], Marko Mutanen [17], Mikko Oranen [17], Riku Paavola [18], Risto Renkonen [19], Pauliina Schiestl-Aalto [20], Mikko Sipilä[20], Maija Sujala [12], Janne Sundell [15], Saana Tepsa[14], Esa-Pekka Tuominen[15], Joni Uusitalo [15], Mikko Vallinmäki [21], Emma Vatka [17], Silja Veikkolainen[12], Phillip C. Watts [1] & David Dunson[2]

Citizen science provides large amounts of biodiversity data. Key challenges in unlocking its full potential include engaging citizens with limited species identification skills and accelerating the transition from data collection to research and monitoring outputs. Here we use a large dataset from Finland to show how even citizens who cannot identify birds themselves can contribute to real-time predictions of avian distributions. This is achieved through a digital twin that combines smartphone-based citizen science with long-term knowledge in a continuously updating model. The app submits raw audio to a backend that classifies birds with machine learning, reducing variation in data quality and enabling validation and reclassification by continuously improving classifiers. We counteracted spatiotemporal sampling biases by interval recordings and permanent point count networks. Over 2 years, the app generated 15 million bird detections. Independent test data show that the digital-twin-informed models are more accurate at predicting bird spatiotemporal distributions. Because our approach is highly scalable and has the potential to generate biomonitoring data even in understudied areas, it could accelerate the flow of reliable biodiversity information and increase inclusivity in citizen science projects.

Biodiversity is integral to maintaining healthy ecosystems and thus to supporting human health, food security, climate stability and agricultural productivity[1–3]. To effectively guide environmental policies and conservation efforts, we need tools that can rapidly and accurately inform about the current and future state of biodiversity[4,5]. Yet, contemporary biodiversity predictions remain inaccurate, particularly at fine spatiotemporal resolutions[6], despite the increasing availability of extensive, long-term biodiversity data[7–10], rapid

advancements of technology to collect large biodiversity data[11,12], and continuously improving modelling tools[13,14]. Reasons why reliable biodiversity prediction has remained so challenging include the inherently complex dynamics of ecological systems[15], the diverse and often inconsistent sources of large datasets[16,17], and the lack of modelling tools capable of rapidly converting the continuous data streams into information transferable to policy and management recommendations[18].

**Fig. 1 | The citizen science smartphone application MK. a**, The app has a continuously updating collective observation board where users can relate their detections to those of the other users (detections exemplified in the map for 1 to 4 April 2025. **b**, The machine-learning-based classifications are calibrated to a probability scale and highlighted with green colour if probability exceeds 0.90. **c**, In collaboration with national parks and municipalities, we implemented 580 permanent point count locations where citizens can make systematic 5-min recordings. **d**, To increase societal impact, user commitment and educational use in Finnish schools, we implemented a bird game through which citizens can learn bird vocalizations. **e**, The aggregated duration of recordings and number of detections per day peak during spring but remain continuous over the entire year. During peak days, the app has accumulated >1,000 h of recordings (with a median length of 33 s) which involve >100,000 detections. **f**, Among the 263 species that can be detected by the app, 110 have been observed with 90% confidence >5,000 times.

As a partial solution to the challenge of achieving fine-resolution biodiversity data, much hope has been invested in the unparalleled potential of citizen science to provide data on a massive scale. Although the potential of citizen science has been repeatedly demonstrated[19–22], data generated by citizen scientists are fraught with sources of biases and noise, potentially compromising the reliability of the resulting inference[23]. Most critical observer-based biases in citizen science relate to heterogeneity in participation, detectability, sampling and preference[24]. As it is difficult to reliably account for the variability in citizens in their skills of identifying species, as well as to quantify the spatiotemporal sampling effort, it remains hard to disentangle biological signals from these observation biases, especially if sampling effort is not carefully documented[25,26].

Digital twinning refers to the concept of creating a digital counterpart of a real-world system. In ecology, digital twinning could mean building a dynamically updated digital model of a species' distribution or an ecosystem's state, based on continuously incoming observational data. While originally developed in engineering to simulate and optimize physical systems[27], digital twinning is gaining interest in biodiversity research, where it can help integrate data, models and expert knowledge in near real time[28–30]. This approach holds promise for improving ecological forecasting and supporting timely environmental decision-making[31]. However, the development of digital twins (DTs) for biodiversity remains a complex and emerging research frontier, hindered by the complexity of natural ecosystems, the need to combine heterogeneous data sources and the technical challenges associated with generating and processing real-time biodiversity data streams.

This Article aims to demonstrate the applicability of DT approaches in biodiversity research for achieving accurate, real-time predictions of species distributions. We illustrate this through a case study in audio-based bird monitoring, showcasing how reliable real-time biodiversity predictions can be achieved through a DT approach that combines the strengths of citizen science, machine learning and high-performance computation. We build on recent approaches in data integration[32,33] and integrated species distribution modelling[34] to combine the continuous flow of new citizen science data with previous long-term data on bird spatial distributions, timing of migration and patterns of singing activity.

Our approach features a continuous model updating process, ensuring that the digital version and its predictions remain responsive to real-time changes in bird activity and environmental conditions. A core feature that distinguishes digital twinning from data integration, is that digital twinning goes further by maintaining a dynamically updated model that mirrors the real-word system as it evolves over time, here distributions, migrations and singing activity of birds, as well as citizens recording them. By relying solely on machine-learning-based bird classifications rather than citizen-based classifications, we remove an important part of observer heterogeneity and increase inclusivity by enabling ordinary citizens without bird identification skills to take part in data collection by making bird recordings. The technological innovations developed in this study not only reduce the time required to generate accurate biodiversity information for policy and management but also increase inclusivity by broadening the stakeholder community and the roles of the stakeholders. This approach empowers and

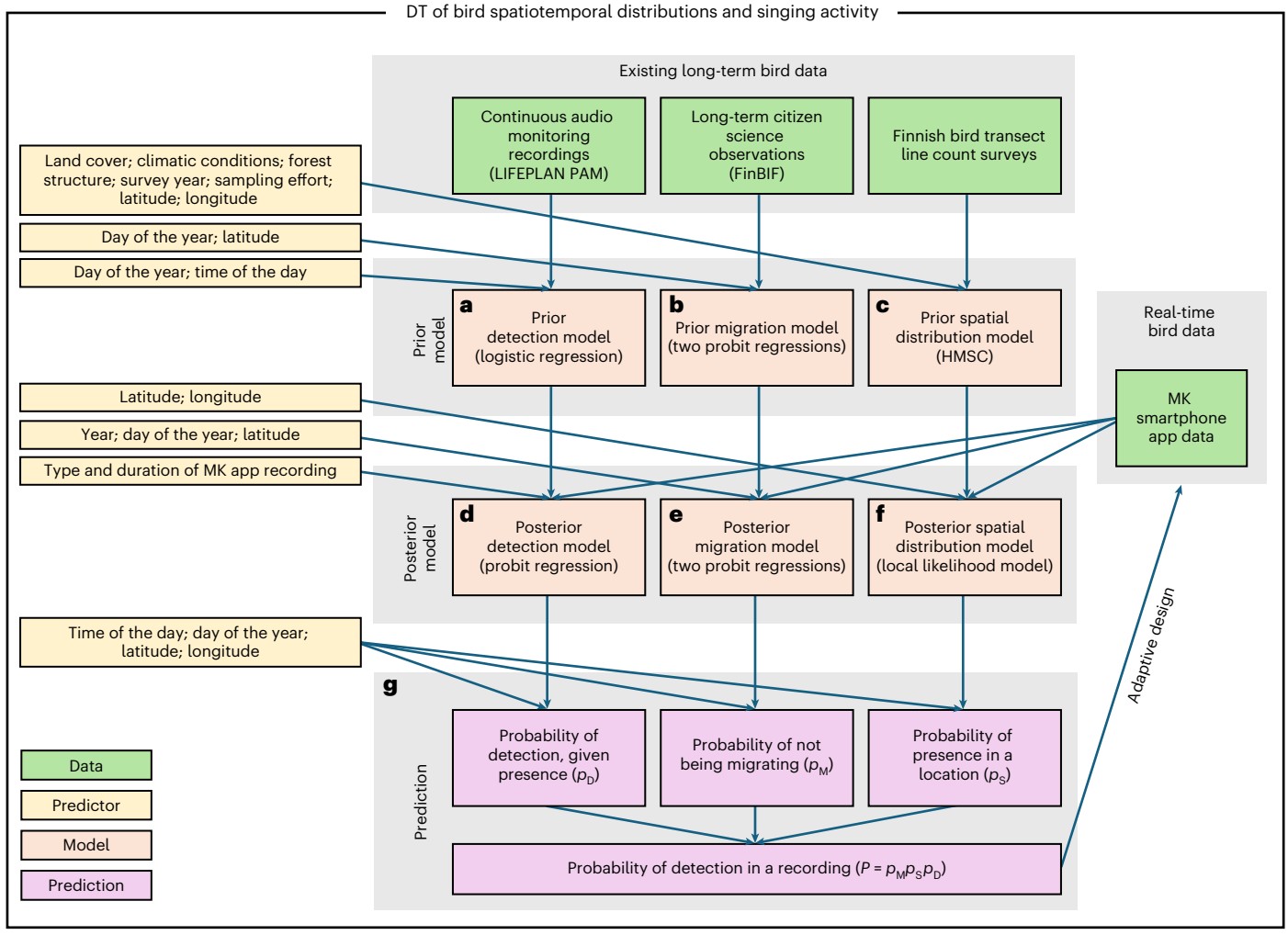

**Fig. 2 | Overview of the DT modelling strategy.** We parameterized a prior model by combining long-term bird observations with spatial and temporal predictors. **a**, Continuous recordings provide prior information about when birds vocalize, conditional on their presence. **b**, Long-term citizen science observations provide prior information about the timing of migration. **c**, Systematic transect line counts, as combined with data on land cover, forest structure and climatic predictors, provide prior information about the spatial distributions of birds. **d**–**f**, The continuously accumulating MK app data are used to update the detection model (**d**), the migration model (**e**) and the spatial distribution model (**f**) and, hence, knowledge of bird spatiotemporal distributions and singing activity. **g**, Probabilistic predictions by the three model components yield the probability that a given bird species is detected in a given MK app recording, as for this to happen (1) the bird should have returned from migration (or be resident), (2) the location should be part of the birds spatial distribution and (3) the bird should vocalize in a manner that leads to detection in the MK app.

engages citizens to provide pivotal contributions to both scientific research and environmental monitoring.

## A tool for digital citizen science

We created a smartphone app called *Muuttolintujen Kevät*, henceforth the MK app, with the Finnish name meaning 'The spring of migratory birds' (Fig. 1). The app was launched on 30 March 2023, through a publicity campaign run in collaboration with the Finnish broadcasting company Yle. The MK app includes three recoding types: (1) direct recordings, (2) interval recordings and (3) point count recordings. The MK app was specifically designed to overcome two critical limitations of citizen science in biodiversity research.

First, to mitigate the differences in species identification skills among citizens, all classifications are performed by a machine learning model, and thus bird identification by citizens is not required. Importantly, not only the classifications but also all raw audio data are submitted and stored in the MK server. This allows the reclassification of the audio with continuously improving machine learning models, as well as the manual validation of species detections if necessary. For this study,

we fine-tuned a baseline BirdNet model[35] for 263 Finnish bird species (all breeding species, non-breeding migrants and most common vagrants) using high-quality annotations generated by bird experts[36]. The model was calibrated specifically for the MK app data, and a 90% confidence score, which we used as threshold for the analyses presented in this Article, can be interpreted as 90% probability of correct classification.

Second, to mitigate spatial observation bias and preferential sampling, the MK app enables not only direct recordings, but also interval recordings and systematic point counts. In the interval recording mode, the app records 1 min every 10 min, continuing up to 12 h. This enables citizens to record, for example, overnight in their yard, including the very early morning hours when birds are most vocal. While the interval recordings do not remove the spatial bias of where the recordings are conducted, they largely remove the temporal preferential bias of when they are conducted. Even if the initiation of an interval recording would be triggered by bird vocalization activity, after the first 9-min break, the recorded minutes represent bird vocalization activity in a much less biased way than direct recordings. The permanent point count network was established in collaboration with Finnish national parks

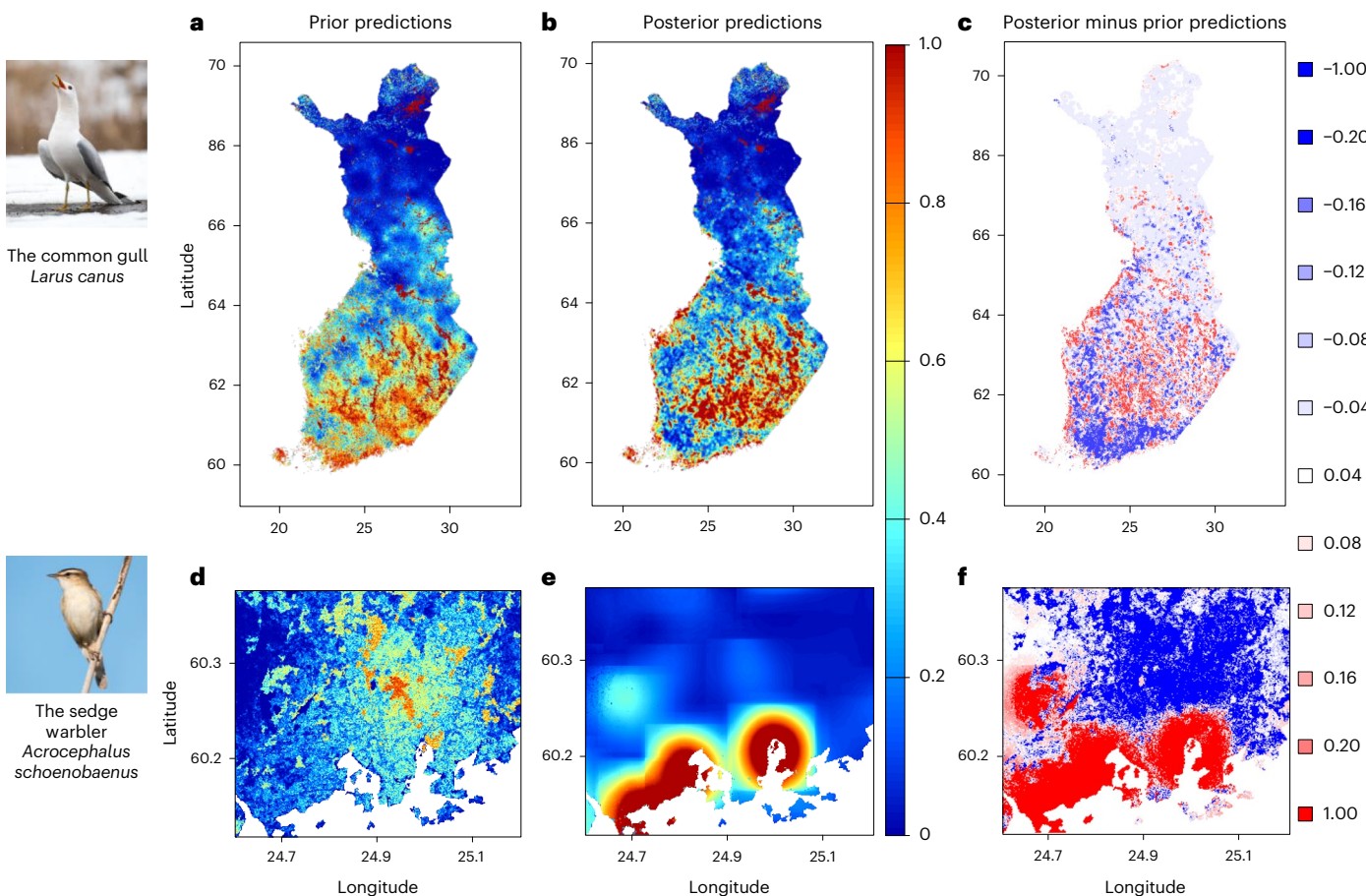

The common gull
*Larus canus*

The sedge warbler
*Acrocephalus schoenobaenus*

**Fig. 3 | Example illustrations comparing posterior and prior predictions of spatial distributions. a–f**, National-level distributions for the common gull (*Larus canus*) (**a**–**c**) and smaller-scale distributions for the sedge warbler (*Acrocephalus schoenobaenus*) around the capital area (**d**–**f**). The prior model is based on long-term bird data only, whereas the posterior model also utilizes observations acquired by digital citizen science through the MK app. The spatial predictions are shown for the prior mean (**a** and **d**), posterior mean (**b** and **e**) and the difference between posterior and prior mean (**c** and **f**).

and municipalities and includes 580 preselected locations in which the citizens can conduct a systematic recording (Fig. 1c). The permanent point count locations mitigate spatial observation bias, as the citizens make recordings at preselected locations. They also partially mitigate the temporal bias, because the recording interval is 5 min long, and thus especially its latter part is less dependent on whether bird vocalization activity triggered the initiation of the recording. We have furthermore encouraged users to initiate point count recordings whenever they walk through the route, disregarding whether birds are vocalizing or not. To engage the users and support their education on bird sound identification, a gamified bird vocalization training feature was added to the MK app in spring 2025 (Fig. 1d).

The MK app rapidly gained popularity among Finnish citizens, with 315,609 individuals (5% of the national population) submitting at least one recording by 29 September 2025. By this date, the app has yielded 16.3 million recordings which contain 15.0 million bird detections with at least 90% classification probability. Most recordings and detections are made through direct recording, but a substantial proportion is also obtained through the interval and point count recordings (Fig. 1e). The detections involve 261 species, out of which 110 have been detected at least 5,000 times (Fig. 1f). In addition, the MK app is used actively for nature education in Finnish schools. For example, a single bird observation event organized on 7 April 2025 was attended by 3,900 school children representing 73 schools. Furthermore, by 13 October 2025, the bird game has attracted 34,248 users, who have together scored a total of 4.2 million identification attempts, each involving the selection of the correct vocalizing species from four candidate species.

## A real-time biodiversity DT

We developed a DT that predicts spatiotemporal distributions of bird occurrences and their vocal activity across Finland, with a spatial resolution of one-hectare, and a temporal updating frequency of one day. The DT operates by updating a prior model each night using the latest data accumulated through the MK app (Fig. 2).

The model predictions are a product of three probabilistic components. First, the migration model yields the probability $p_M$ by which the species is present from the point of view of their migratory behaviour (with $p_M = 1$ for non-migratory species), given the latitude, year and the day of the year. Second, the spatial distribution model yields the probability $p_S$ by which a given location is part of the species distribution during the non-migratory period. Third, the detection model yields the probability $p_D$ by which, conditional on a species being present, it vocalizes in a way that the MK app detects it with at least 90% classification probability. The detection model is parameterized in terms of the day of the year, time of the day, and the length and type of recording. The product of these three probabilities ($p = p_M p_S p_D$) yields the probability the species is observed in a given MK app recording.

We inferred the prior migration model using long-term citizen science data on species observations (Fig. 2b). We quantified prior knowledge on bird species' spatial distributions by fitting the joint species distribution model Hierarchical Modelling of Species Communities (HMSC)[37] to long-term data on transect-line surveys, using as predictors 1-ha-resolution raster maps of land-cover variables, forest structure variables and climatic variables (Fig. 2c). The prior detection model was inferred using 4-year-long continuous passive audio

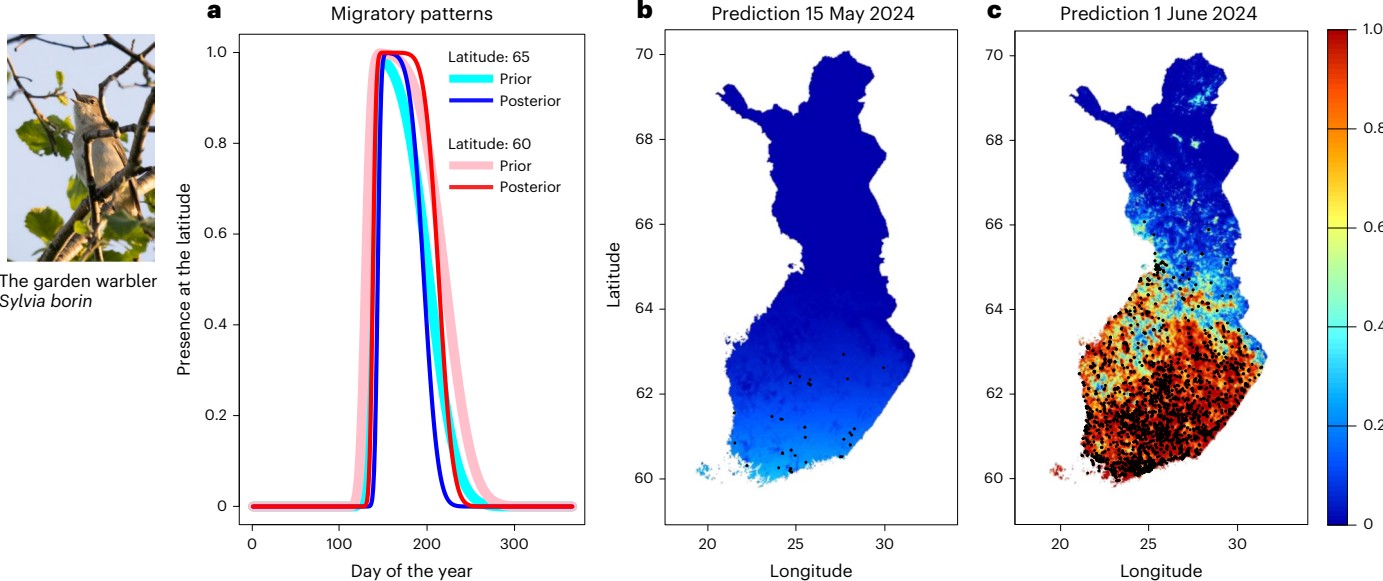

**Fig. 4 | Example illustrations of spatiotemporal predictions for the garden warbler (*Sylvia borin*). a**, Comparison of migratory timing between the prior model and the posterior model for 2024. **b,c**, Posterior predictions for the species distribution in the beginning (**b**; 15 May) and in the end (**c**; 1 June) of the realized migratory period in 2024. The black dots in **b** and **c** show the MK app detections for the exemplified days.

monitoring (PAM) data from seven Finnish research stations[38]. We used logistic regression to model the probability by which a passive audio recorder would detect a vocalization of a given species as a function of the day of the year and time of the day, conditional on the species being present at the location (Fig. 2a).

We used the MK app data to update the migration functions and spatial distributions at daily intervals. This is a computationally intensive task, which we simplified by modelling species independently and updating the prior in stages. The first stage is to translate the detection model from PAM to the MK app data; this is achieved using a probit model that accounts for the length and type (direct, interval or point count) of the recording. The second stage updates the parameters of the migration model using MK app data directly, with the overall shape of the migration probability curve over time shrunk to the prior using a functional penalty to promote stability and improve forecasting. Finally, the spatial distribution is updated using a local-likelihood method, in which each cell is updated based solely on nearby data. This approach allows cells to be updated in parallel and is critical for scalability. The updating of the spatial distribution component is conducted directly at the level of the model predictions through spatial smoothing, not at the level of prior model parameters that map, for instance, the environmental affinities of the species. Full details are available in the Methods.

## Example predictions

The DT continuously updates its long-term knowledge on bird spatiotemporal distributions through the newly accumulating citizen science data. As illustrated in Fig. 3 for two example species, the posterior predictions of species distributions can deviate substantially from the prior predictions both at large and small spatial scales. This indicates that the DT undergoes substantial learning. For the common gull (*Larus canus*), the DT increases the contrast between high-prevalence areas (lakes and coastal areas) and low-prevalence areas, thus changing the predictions consistently over large spatial scales (Fig. 3c). For the sedge warbler (*Acrocephalus schoenobaenus*), the posterior predictions deviate substantially from the prior predictions at higher spatial resolution, as illustrated for the capital area in Fig. 3d–f. Sedge warblers breed in reedbeds, which are not well represented in the transect line data and which are not distinguished in the habitat classification used to make

prior predictions. This leads to poor predictive performance of the prior model, leaving room for substantial improvement by the DT in areas with abundant MK app data such as near the capital. The DT also learns to predict bird temporal dynamics. By tracking the daily arrival of migrants, the DT can accurately infer the timing of spring migration and the associated spatial dispersal (Fig. 4a). This results in highly dynamic spatiotemporal distributions, such as for the garden warbler (*Sylvia borin*), where the distribution changes from almost universal absence to widespread presence within 2 weeks (Fig. 4b,c).

## Evaluation of predictive capacity

We evaluated the predictive capacity of the DT with two different test datasets: MK app recordings for the next day, and manual point counts for the next day. For both test datasets, the DT approach substantially improved predictive capacity, as compared with the prior model (Fig. 5). We performed both evaluations for those 89 species for which the MK app data contained at least 5,000 detections in 2024. For both evaluations, we updated the DT model using the MK app data up to the previous day and then used the test data to evaluate the predictions of both the prior and the DT models.

For the evaluation against future MK app data, we used the year 2024 as the evaluation period. Based on the location, time, type and duration of each MK app recording, we predicted detection probabilities for each species by both the prior and the DT models, with the DT incorporating data up to the previous day (Fig. 5a). The DT substantially improved the next-day MK app predictions for bird detections, with the mean area under the curve (AUC) across 89 species increasing from 0.71 to 0.77. The improvement was most pronounced for migratory species and for species with initially poor prior model predictions (Fig. 5b,c).

To further evaluate the difference between the posterior and prior predictions against fully independent data, we performed manual point counts by bird experts in preselected locations from 7 May to 7 June 2025. The bird experts were seasoned volunteer birdwatchers, whose capacity to identify birds from their vocalization has been demonstrated, for example, by providing high-quality survey data to the national line transect or point counting schemes. The manual point count locations were selected algorithmically to represent different combinations of prior and DT predictive probabilities, prioritizing sites where the prior and posterior predictions were most contradictory.

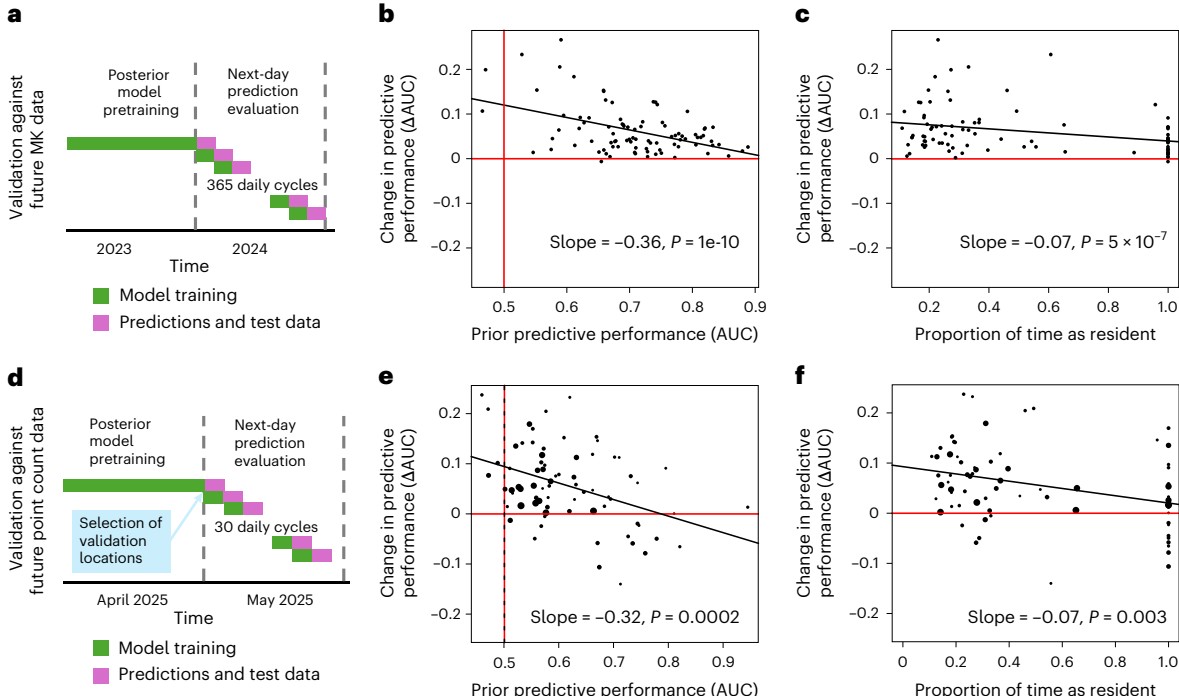

**Fig. 5 | Comparison of DT (posterior) and prior models in terms of predictive power. a–c**, Evaluation of the capacity to predict future MK app data. We updated the DT dynamically until the present day (**a**) and contrasted the next day's prior and posterior predictions to actual MK data (**b** and **c**). The results are shown for those 89 species that were observed at least 5,000 times by the end of 2024. **d–f**, Evaluation of the capacity to predict future point count data by experts. We used the DT predictions from the end of April 2025 to select point count locations that showed the greatest differences between the prior and DT predictions (**a**), and then compared the next day's prior and posterior predictions with the actual point count data (**e** and **f**). The results are shown for those 73 species that were observed at least 10 times in the expert point counts. The dot size in **e** and **f** is proportional to $p(1 - p)$, where $p$ is the species' prevalence in the point count data and, hence, larger dots show cases where the AUC can be calculated more reliably. The DT makes substantially better predictions for most species (mean difference in AUC between posterior and prior predictions 0.06 for both types of prediction), and especially for species for which the prior model is poor (**b** and **d**) and that are migratory (**c** and **f**). The *P* values and slopes in **b**, **c**, **d** and **f** originate from a linear model that includes as explanatory variables both prior predictive performance and the proportion of time that the species spends as a resident.

These locations were visited by bird experts, who performed a total of 1,185 5-min point counts without knowing the prior and DT predictions by which the locations were selected. This test confirmed that the DT leads to improved predictions: the mean AUC across those 73 species that were available for this comparison increased from 0.62 to 0.67 for the expert point count data. This was again the case especially for species for which the prior model was poor (Fig. 5e) and that are migratory (Fig. 5f).

We further compared the DT predictions with those based on the eBird[39] global citizen science project. For each survey week, we extracted species occurrence probabilities from the eBird Status and Trends Weekly Abundance Maps released in summer 2025, which represent data accumulated through 2023[40]. For those 53 species for which eBird-based predictions were available, the mean AUC was 0.62 for eBird-based predictions and 0.67 for DT predictions.

## Discussion

The need for accurate and real-time biodiversity predictions has been much advocated[14,15,41], but achieving this has remained challenging[28]. This Article demonstrates the feasibility of constructing a real-time DT of biodiversity and shows, with independent test data, that the DT improves predictions about the current and future states of biodiversity. By combining long-term data with a continuous stream of smartphone-based citizen science data, our DT addresses the challenge of generating reliable predictions at a high spatiotemporal resolution[6]. Such predictions are much needed under the UN Convention on Biological Diversity's Global Biodiversity Framework to detect biodiversity changes and to promptly implement the necessary environmental

management and policy actions. Although the DT developed here is aimed at quantifying changes in species distributions rather than directly identifying their potential drivers or recommending management or policy actions, it provides a foundation for making informed progress in these directions.

Citizen science can provide massive amounts of biodiversity data. For example, the platforms eBird[39], iNaturalist[42] and Pl@ntNet[43] have recruited some 1.1 million, 8.9 million and 8.2 million users, respectively. These extensive citizen science datasets have not only provided an invaluable resource for biodiversity research but have also stimulated the development of numerous statistical methods to address data quality issues, such as sampling biases and detection errors[23]. For example, although eBird's data collection procedures involve systematic quality control and quantification of user skills, using these data for prediction and inference requires statistical approaches that carefully account for confounding factors and changes in the observation process. The best practice recommendations for using eBird data involve choices related to filtering the data for complete checklists, performing spatial subsampling and using filters for observation effort[44]. The predictions based on eBird data that we utilized in our comparison are not updated automatically in real time, but periodically by Cornell Lab of Ornithology data scientists, who provide Status and Trends products based on data accumulated over several years.

A core feature of the MK app is that it was directly developed to overcome the outstanding challenges of citizen science[23]. First, to tackle the issue of variable and often unknown sampling effort, the MK app quantifies the location, time, type and duration of each recording and implements standardized interval recordings and permanent point

count routes. Second, to remove observer heterogeneity in species identification, the MK app uses machine-learning-based classifications with well-calibrated estimates of uncertainty. These characteristics of the MK app data facilitated their straightforward integration into a predictive DT approach. Furthermore, by storing raw audio data, the DT enables reclassification of past observations with continuously improving classification models, ensuring that they remain useful and accurate over time. Despite the above-mentioned features, the MK app data have some of the biases that are characteristic to citizen science datasets. Most importantly, the direct recordings are triggered by bird vocalizations that are of interest to the users. As shown in our previous analysis, some users target only new species that they have not recorded before, whereas other users provide data that are comparable to PAM[45]. Accounting for such variation in user profiles provides an important challenge for future work. Another limitation is that the MK app is based on audio only, omitting visual observations of birds.

Global biodiversity databases have major spatial biases, which influence our understanding of biodiversity and hamper its protection[46]. Many areas remain understudied due to barriers related to wealth, language, geographical location and security[47], making it difficult to implement large-scale biomonitoring programmes that would require transport of specialized experts and equipment at appropriate times. To fill such spatial and temporal gaps in biodiversity data, the potential of citizen science has been previously recognized[48]. Our approach is highly scalable both computationally and in terms of smartphone technology and can be easily extended across geographical regions, given the widespread global ownership of smartphones. Even for birds, one of the best-studied taxonomic groups, and in Finland, a country with exceptionally well-documented biodiversity, our DT approach demonstrated substantial improvements in predictive performance. Thus, in regions where biodiversity is less well studied, our technology offers strong potential to rapidly improve ecological knowledge and inform conservation efforts.

Our DT approach may not generalize straightforwardly to many existing citizen science data streams, as the seamless integration between the data collection and the real-time predictive modelling was enabled by the fact that the MK app was specifically designed to serve this purpose. While building an operational DT such as the one presented here may initially require more effort than most other citizen science platforms, its capabilities go well beyond what static systems can achieve, as it provides a dynamic approach for forecasting biodiversity. Compared with this effort, the improvement in predictive power that we reported here may appear moderate: the AUC improved from 0.62 in our prior model to 0.67 in the DT. However, we argue that this improvement is substantial, as the AUC value increased by 42% if compared with the baseline value of 0.50. Instead, the low AUC values are explained by the fact that the predictive task that we targeted is highly challenging. Namely, our test data concern variation in species detections over a small geographic area (where all the species generally occur) and over a short period (during which all the species were generally present), making it highly challenging to predict in which samples the species were present and in which they were absent.

Successfully protecting nature requires collaboration between governments, businesses and civil society, with a key question being how to engage a larger part of society in supporting nature[49]. Citizen science has great potential to engage people more actively in environmental monitoring[50], and our DT addresses major challenges associated with large-scale spatial monitoring in citizen science[51]. Moreover, mobile-based approaches may attract younger participants, making them an important means of increasing public understanding of science[52]. Thus, these approaches can be effective in motivating participants to sample biodiversity in more meaningful ways, potentially reducing some of the biases inherent in how citizen science data are collected[53]. The MK app has substantially promoted citizen engagement and helped reconnect citizens with nature through extensive school collaboration, media coverage, the possibility of sharing results through social media, and educational features such as the bird game. In particular, the MK app has gained popularity among ordinary citizens who do not necessarily recognize any bird sounds themselves, as it enables them not only to learn which birds vocalize in their surroundings, but also to contribute valuable biodiversity data that are immediately used for research and monitoring. This inclusivity, together with the DT approach, greatly enhances the ability of citizen science to provide reliable, real-time information on global biodiversity, helping to bridge the current time lag between research and policy.

## Methods

### The MK smartphone app

The MK mobile application and its technology infrastructure were developed collaboratively by the University of Jyväskylä, CSC – IT Center for Science and the University of Helsinki. The MK app is built upon an open-source technology stack and developed using the Flutter mobile application framework. The MK app is freely available on Android and Apple mobile devices in Finland and Sweden. At the core of the application's architecture lies a server-side, customized Camunda BPM hyperautomation platform, providing a robust solution for anonymous user participation in research and data collection processes. This architecture addresses challenges related to European Union data protection regulations (General Data Protection Regulation), ensuring secure application use and enabling the transfer of data for research purposes.

The operational workflow is initiated by a user recording bird sound—including direct, interval or point count recordings—along with metadata such as an anonymous participation key, location, recording length and timestamp. These data are transmitted via Internet connection to an application programming interface running within a secure computing environment provided by CSC. The audio files are stored in CSC's object storage system Allas, while the metadata are saved in a MongoDB database. The workflow directs the audio files to several virtual machines running in the cPouta cloud service, which performs bird classifications. The results are returned to the user, who can voluntarily assess the correctness of the identifications and provide feedback to further develop the classification model. The backend stores observation data and results for scientific purposes and retains the original audio files, allowing reprocessing with future classification models and manual validation of observations.

### The machine-learning-based model for bird classification

The bird species classifications are produced with a convolutional neural network that consists of a pretrained convolutional base of EfficientNet B0 architecture from BirdNET-Analyzer[35] and a classification head that we fine-tuned with vocalizations of 263 Finnish bird species[36]. Although the list of the selected 263 species is not the full list of all 496 species ever recorded in Finland, it contains all breeding species, non-breeding migrants and most common vagrants, making it unlikely that a citizen records a species not included in the classification model. The training dataset combined targeted recordings from Xeno-canto[54], soundscape recordings from eight sampling locations in Finland, targeted field recordings by Harry J. Lehto and selected mobile phone recordings produced by MK app users. Labels for training data were collected through Bird Sounds Global annotation portal (https://bsg.laji.fi).

The classification model analyses the recordings in 3-s segments. The audio signal is converted into spectrogram images with an overlap of 1 s between consecutive segments using short-time Fourier transform. For each segment, the model predicts detection probabilities for all species. The model predictions were calibrated with species-specific logistic regression models. We selected 80 vocalizations per species from the MK phone recordings uniformly across confidence bins ranging from 0.2 to 1.0. The binary labels (presence/absence of the species)

were provided by a bird expert who listened to the recordings. The predictions of highly unlikely species are penalized on the basis of location and day of the year to remove obvious misclassifications (for example, migratory species detected during winter) from the data.

## The citizen science campaign

The MK application was launched in collaboration with Finland's national public broadcasting company Yle, which substantially amplified its visibility in national media. The first public mention occurred on 12 April 2023, during the *Metsäradio* ('Forest radio' in Finnish) programme, which focuses on forestry, nature and outdoor lifestyle. Subsequent coverage included the *Luontoilta* ('Nature evening' in Finnish) radio broadcast on 4 May, and a featured theme on Yle's special television programme *Muuttolintujen Kevät* ('Spring of migratory birds' in Finnish) on 10 May. The application was also highlighted in Yle's main evening news broadcast, which reaches an average television audience of approximately 750,000 viewers—roughly 14% of the Finnish population. In addition to traditional media, Yle promoted the citizen science campaign through its social media channels throughout the spring. Simultaneously with Yle's campaign, the University of Jyväskylä organized citizen science events for a local audience during three consecutive springs. During these events, citizens had the possibility to interact with scientists about topics related to the MK application, and more broadly about birds and environmental change. The MK app and the citizen science events received attention in several local and national newspapers, as well as in birdwatching-related communities such as local BirdLife partners.

In spring 2024, we published educational material to help teachers integrate the MK app into their teaching. The material, which supports Finland's national curriculum for basic education, is freely available in Finnish and Swedish (https://mappa.fi/materiaalit/muuttolintujen-kevat-sovellus). This material was developed in collaboration with the Finnish Association of Nature and Environmental Schools and the Central Finland LUMA Centre. The educational use of the MK app was tested nationally in the *Suuri Linturetki* (Great Bird Excursion) event aimed at primary schools, as well as in the LUMA Centre Finland's remote afternoon club for 1st and 2nd graders.

## Overview of the DT approach

We index the MK app recordings by $i = 1, …, n$, where $n$ equals 16.3 million by 29 September 2025. Each recording is characterized by its time $t_i$ (containing year, day and time of the day with 1 s resolution), location $c_i$ (latitude and longitude, or alternatively the corresponding index of the one hectare cell of the spatial grid for which we predicted species distributions), recording type $r_i \in \{\text{direct, interval, point}\}$ and log-duration $x_i$. For each recording $i$, we denote the bird detections by $Y_{ij}$, where $j = 1, …, p$, with the total number of bird species $p$ that may be detected equalling 263. We define $Y_{ij} = 1$ if bird species $j$ was detected in the recording, in the sense of the classification model predicting detection with at least 90% probability, and $Y_{ij} = 0$ if this is not the case.

Under the prior model, the probability of detection factors as

$$P[Y_{ij} = 1] = m_j(c_i, t_i | Y^m)\, s_j(c_i | Y^s, X^s)\, d_j^{\text{PAM}}(t_i | Y^d),$$

where the three components relate to migration ($m$), spatial distribution ($s$) and detection ($d$). The migration model yields the probability $m_j$ of the species $j$ being present, from the migration point of view, at time $t_i$ and latitude $c_i$. The prior migration model is fitted using long-term citizen science data $Y^m$ on species observations. The spatial distribution model yields the probability $s_j$ of the species $j$ being present at the grid cell $c_i$, conditional on it being present from the migration point of view. The prior version of the spatial distribution model is fitted using long-term systematic bird transect count data $Y^s$, as well as predictors $X^s$ related to habitat and climatic conditions. The detection model yields the probability $d_j$ by which the bird is detected

at time $t_i$ by the classification model with at least 90% probability, conditional on the species being present from both the migration and spatial distribution points of view. The detection model is parameterized with long-term continuous PAM data $Y^d$. Consequently, $d_j^{\text{PAM}}$ models the probability that a bird would be observed in a 1-min-long PAM recording rather than in an MK app recording, as indicated by the superscript PAM. We note that the detection model still contains useful information about how bird vocalization activity (hence, detection) depends on the day of the year and the time of the day.

In the posterior model of the DT, the probability of detection is modelled as

$$P[Y_{ij} = 1] = m_j(c_i, t_i | Y^m, Y^{\text{MK}})\, s_j(c_i | Y^s, X^s, Y^{\text{MK}})\, d_j^{\text{MK}}(t_i, r_i, x_i | Y^d, Y^{\text{MK}}),$$

where all the three model components are updated by the MK phone app data $Y^{\text{MK}}$. In addition, the detection model is transferred from probability of detection by 1-min-long PAM ($d_j^{\text{PAM}}$) to probability of detection by MK phone app recording ($d_j^{\text{MK}}$). This brings dependency on the type ($r_i$) and log-duration ($x_i$) of the recording.

We next describe how each component of the prior model was inferred, and then how the DT updates the posterior by the MK app recordings.

## Prior model for detection

The prior detection model yields the probability $d_j^{\text{PAM}}(t_i | Y^d)$ by which the bird is detected at time $t_i$ from 1 min of PAM recording by the classification model with at least 90% probability, conditional on the species being present from both the migration and spatial distribution points of view. To parameterize the detection model, we utilized 595,400 1-min-long recordings acquired from 26 January 2021 to 10 February 2023 in eight Finnish sites that took part in the LIFEPLAN biodiversity sampling scheme[38]: Värriö Subarctic Research Station, Hyytiälä Forest Station, Konnevesi Research Station, Lammi Biological Station, Kiiminki Field Site, Archipelago Research Institute, Oulanka Research Station and Kilpisjärvi Biological Station. Using the above-described bird classification model, we inferred the presence–absence of Finnish birds in each 1-min segment, using as classification threshold 0.75, or the 90% quantile of all classification probabilities. To model detection conditional on the species being present at the site, we selected for each year and each site the time period that started after 5% of all the detections for that year had accumulated and ended when 95% of all the detections for that year had accumulated. We modelled the data with logistic regression (function glm in R with binomial family) using the time of the year (number of days since 1 January) and time of the day (minutes from midnight) as predictors. We modelled the seasonal and diurnal effects through the periodic functions of $\sin(2\pi x)$, $\cos(2\pi x)$, $\sin(4\pi x)$ and $\cos(4\pi x)$, where $x$ represents either the day of the year or the time of the day, both scaled to the range 0–1. We fitted the models to those 117 species for which the LIFEPLAN data were sufficient. We validated all the detection models by bird experts. If a bird expert considered that the detection model poorly reflected the species' actual vocalization activity pattern, we manually adjusted the parameters to better align with expert judgement. For the remaining 146 species for which the LIFEPLAN data were not sufficient for statistical model fitting, we parameterized the detection models solely on the basis of expert elicitation.

## Prior model for migration

The prior migration model $m_j(c_i, t_i | Y^m)$ yields the probability by which the species is present from the point of view of migratory behaviour at the location $c_i$ and time $t_i$. We parameterized the migration model as $m_j(c_i, t_i) = \min \left\{ \Phi\left[\text{day}(t_i); \mu = \theta_j^{S,M} + \theta_j^{S,L}\text{lat}(c_i), \sigma = \theta_j^{S,I}\right], 1 - \Phi\left[\text{day}(t_i); \mu = \theta_j^{A,M} + \theta_j^{A,L}\text{lat}(c_i), \sigma = \theta_j^{A,I}\right]\right\}$ with $\Phi$ denoting the cumulative density function of the normal distribution with mean $\mu$ and standard deviation $\sigma$, lat($c_i$) the latitude of location $c_i$, and day($t_i$) $\in \{1, …, 365\}$ the day

associated with time $t_i$. Therefore, our prior migration model is not specific to any given year but seeks to capture the averaged migration timing, whereas the posterior model yields year-specific predictions. Note that $\Phi$ increases from zero to 1 as its argument increases. The first part of the function models the arrival of the species during spring migration: $\theta_j^{S,M}$ models the mean day of arrival, $\theta_j^{S,L}$ models its latitude dependence and $\theta_j^{S,I}$ characterizes the length of the interval during which arrival occurs. The second part of the function models the departure of the species during autumn migration: $\theta_j^{A,M}$ models the mean day of departure, $\theta_j^{A,L}$ models its latitude dependence and $\theta_j^{A,I}$ characterizes the length of the interval during which departure occurs. The migratory behaviour of each species $j$ is thus captured through six parameters that we combine into the vector $\theta_j = (\theta_j^{S,M}, \theta_j^{S,L}, \theta_j^{S,I}, \theta_j^{A,M}, \theta_j^{A,L}, \theta_j^{A,I})$.

To estimate $\theta_j$, we downloaded from Finnish Biodiversity Information Facility (FinBIF; https://laji.fi/en) citizen science observations $Y^m$ on Finnish birds for the period 2000–2022. For each year, each species and each of ten evenly distributed latitude zones, we defined the 'first' and 'last' observation as the 5% and 95% quantile of observations for which at least 50 occurrences were available. We fitted linear models (with function lm in R) to these data, with latitude as the sole predictor, yielding estimates of $\theta_j^{S,M}$ and $\theta_j^{S,L}$ when using first observations as the response, and estimates of $\theta_j^{A,M}$ and $\theta_j^{A,L}$ when using last observations as the response. To parameterize the lengths of the intervals during which the spring and autumn migrations progress, we calculated (with function predict in R) the upper and lower 95% prediction intervals for each latitude zone, and then defined the parameters $\theta_j^{S,I}$ and $\theta_j^{A,I}$ as one-fourth of the difference between the upper and lower intervals, averaged over the latitude zones. The FinBIF data enabled fitting the migration model for 160 species. As citizen science data are prone to errors, we manually validated all the migration models by bird experts. We filtered out clearly erroneous observations in the FinBIF data and manually adjusted parameters as needed to better reflect expert opinion on migratory timing. For the remaining 103 species for which the FinBIF data were not sufficient for statistical analyses, we inferred the migration model parameters using expert elicitation.

### Prior model for spatial distribution

The prior spatial distribution model yields the probability $s_j(c_i|Y^s, X^s)$ by which each species $j$ is present at each grid cell $c_i$, conditional on it being present from the migration point of view. We predicted these species occurrence probabilities for Finland at 1-ha resolution with the joint species distribution modelling approach HMSC[37].

As the response data $Y^s$, we used expert-based Finnish bird transect line count surveys conducted during 2006–2023. The data consist of 4,014 surveys on 555 different routes systematically covering the whole country with 25-km intervals. Each survey route contained counts on birds observed during a 6-km-long transect. We included in the HMSC analysis those 137 species that were observed in at least 100 surveys. We converted the prevalences into presence–absence data and assumed Bernoulli distribution with a probit link function.

As predictors $X^s$, we used variables representing land cover, climatic conditions, forest structure, temporal trends and sampling effort. The land cover variables were derived from the CORINE land cover database for 2006[55], 2012[56] and 2018[57] and represented proportions in the following categories along the transect: (1) mixed forests, (2) deciduous forests, (3) shrubs, (4) grasslands and wetlands, (5) agricultural land, (6) barren land, (7) urban areas, (8) water bodies and (9) coastal habitats. The climatic variables were derived from Copernicus Climate Change Service[58] and represented the average (10) summer (June and July), (11) winter (December, January and February) and (12) spring (April and May) temperatures during the year before the survey, all included as second-order polynomials. The forest structure variables were derived from the Finnish Multi-Source National Forest Inventory raster maps of the years 2013, 2015, 2017, 2019 and 2021 provided by the Natural Resources Institute Finland[59,60]. These data

represent (13) the stand age, the volumes of (14) pine, (15) spruce, (16) birch and (17) other deciduous trees, as well as (18–22) the five principal components of a detailed categorization to forest types. Temporal trends not explained by the previously mentioned predictors were modelled by including (23) the linear effect of the survey year. Sampling effort was accounted for by including (24) the length of the transect line and (25) the duration of the survey as predictors. We further modelled spatial variation not captured by the fixed effects by including the transect as a spatially structured random effect[61].

We fitted the HMSC model using a Bayesian approach. We used default prior distributions[62] and sampled the posterior distribution using the high-performance computing extension[63] of the R package Hmsc[64]. We sampled the posterior distribution with four chains, ignoring the first 12,500 to allow convergence and thinning the remaining by 100 to obtain 250 posterior samples per chain and, thus, 1,000 posterior samples in total. When using the fitted model to predict the distributions of birds at one hectare resolution, we set all time-dependent predictors to correspond to the year of 2023 and fixed the sampling effort variables to the mean values over the data. We validated all the predicted spatial distributions by bird experts. If a bird expert judged that the spatial distribution poorly represented the species' actual distribution, we applied manual corrections. For the remaining 99 species for which the transect line data were not sufficient for statistical model fitting, we constructed prior models of spatial distributions based on expert elicitation.

### Posterior model for detection

PAM recordings differ in many senses from MK app observations, which are overwhelmingly short opportunistic recordings. We corrected this mismatch by fitting a model that translates the PAM detection probabilities to MK detection probabilities. For computational simplicity, we left the migration model and spatial distributions fixed at their prior values when updating the detection model. That is, we fitted the model

$$P[Y_{ij} = 1] = m_j(c_i, t_i|Y^m)\, s_j(c_i|Y^s, X^s)\, d_j^{MK}(t_i, r_i, x_i|Y^d, Y^{MK}).$$

We parameterized the updated MK detection probabilities with a probit model,

$$d_j^{MK}(t_i, r_i, x_i|Y^d, Y^{MK}) = \Phi\left(\alpha_j + \beta_{j,r_i} x_i + \gamma_j \Phi^{-1}\left(d_j^{PAM}(t_i|Y^d)\right)\right).$$

Here, $\alpha_j$ is a species-specific intercept that accounts for differences in the average number of observations between PAM and MK observations, $\beta_{j,r_i}$ is coefficient of log-duration for recording type $r_i$ and allows the model to adjust for the fact that longer (direct) recordings are more likely to have MK detections, and $\gamma_j$ is coefficient of the linearized PAM detection probability that allows the posterior model to inherit time dependence. We placed independent $N(0, 5^2)$ priors on all coefficients. We obtained maximum a posteriori (MAP) estimates via gradient descent and used these as a plug-in estimate for future inference. Because all MK observations are informative about the detection model, after the first year the posterior variance for estimated parameters is already vanishingly small and refitting the model daily with streaming data does not change estimates or performance to any relevant degree. Consequently, we chose to update the detection model once at the start of each year with all available data.

### Posterior model for migration

Migration within a given year can substantially deviate from the long-term average behaviour captured by the prior. The next step in our analysis was to update the prior migration parameters $\theta_j$ to year-specific posterior parameters $\bar{\theta}_j$. As with the detection model, we fixed the spatial distribution at the prior and fitted the model

$$P[Y_{ij} = 1] = m_{\bar{\theta}_j}(c_i, t_i|Y^m, Y^{MK})\, s_j(c_i|Y^s, X^s)\, d_j^{MK}(t_i, r_i, x_i|Y^d, Y^{MK}),$$

where we temporarily write $m_{\tilde{\theta}_j}$ instead of $m_j$ to make dependence on the parameters clear. Shrinking the posterior migration function towards the prior migration function is key for model stability and accurate forecasting. Specifying a suitable prior is complicated by the fact that parameters are on very different scales, and sometimes large changes in parameters are needed to produce comparatively small changes in the migration function—for example, to produce a step-function pattern in the case of rapid migration. To overcome these challenges, we used a functional prior,

$$p\left(\tilde{\theta}_j\right) \propto \exp\left(-\lambda f\left(\tilde{\theta}_j\right)\right),$$

with $\lambda > 0$ a precision parameter and $f$ the penalty

$$f(\tilde{\theta}_j) = \int \left[m_{\tilde{\theta}_j}(c,t) - m_{\theta_j}(c,t)\right]^2 \mathrm{d}c\mathrm{d}t.$$

This prior assigns high probability mass to migration functions that are close in shape to the prior function, allowing large absolute changes in parameter values while ensuring the general form of the migration function is stable. A large value of $\lambda$ shrinks the posterior migration function more strongly towards the prior migration function. We fixed the relatively small value of $\lambda = 0.01$ and approximated the integral in $f$ over a fine grid. We calculated MAP estimates with gradient descent and used these as a plug-in estimate for forecasting and future model fitting. We designed our migration models to be year specific; therefore, their parameters were reset to the prior at the beginning of each new year and are informed only by data from the particular year. In the next section, we write $m_j(c_i, t_i | Y^m, Y^{MK})$ for the migration model with updated parameters.

## Posterior model for spatial distribution

We next describe how the spatial component of the DT was updated. Adjacent cells are likely to have similar spatial probabilities, but the massive number of 1-ha cells across Finland and need for daily updating make exact Bayesian inference (for example, calculating the posterior with a Gaussian process prior for the spatial component) intractable. We adopted a local-likelihood approach, which estimates the spatial distribution independently in each cell using a weighted log-likelihood that incorporates nearby observations, discounted by their distance. The local log-likelihood for the posterior spatial probability $s$ in a focal cell $c_0$ has the form

$$\ell(s) = \sum_c K(c, c_o) \sum_{i:c_i=c} \left[ Y_{ij}^{MK} \log\left(p_{ij}(s)\right) + \left(1 - Y_{ij}^{MK}\right) \log\left(1 - p_{ij}(s)\right) \right],$$

where $p_{ij}(s) = m_j\left(c_i, t_i | Y^m, Y^{MK}\right) \times s \times d_j^{MK}\left(t_i, r_i, x_i | Y^d, Y^{MK}\right)$, and $K(c, c_0) = \exp\left(-\tau \mathrm{dist}(c, c_0)^2/2\right)$ is a Gaussian kernel. The parameter $\tau$ controls the influence of nearby cells for inferring the posterior probability in $c_0$. If $\tau$ is very large, then the weights assigned to nearby cells vanish and only information in the focal cell is used to update the spatial probability. Conversely, as $\tau$ becomes very small, all cells across Finland are given equal weight when learning the updated spatial probability. In the special case of $\tau = 0$, the local likelihood reduces to the traditional likelihood

$$\ell(s) = \sum_c \sum_{i:c_i=c} \left[ Y_{ij}^{MK} \log\left(p_{ij}(s)\right) + \left(1 - Y_{ij}^{MK}\right) \log\left(1 - p_{ij}(s)\right) \right].$$

In our computations, we used $\tau = 1/2.5^2$, which allows cells within roughly 7.5 km of the focal cell to influence the posterior probability. We used a truncated Gaussian prior for $s$ with moments taken from the prior model: $s \sim N_{0,1}(s_j(c_0 | Y^s, X^s), \sigma_j(c_0 | Y^s, X^s))$. We obtained MAP estimates $\hat{s}(c_0)$ independently for each cell $c_0$ and used these as plug-in estimates to define the posterior model, $s_j(c_0 | Y^s, X^s, Y^{MK}) = \hat{s}(c_0)$. We estimated posterior variances using a Laplace approximation.

To facilitate computations, we defined a downsampled grid at a resolution of 1 km², with each downsampled cell $c'$ containing 100 cells from the original grid. We calculated prior means and variances for each downsampled cell $c'$ by averaging the prior means and variances of the associated 1-ha cells. The local likelihood approach was used to find posterior probabilities for each downsampled cell. To facilitate a fair comparison with the 1-ha prior, we then upsampled the low-resolution predictions. Our method of upsampling aims to preserve the local geometry of the prior with the constraint that average probabilities within each 1-km² cell must be consistent with the downsampled posterior. Accordingly, for each 1-km² cell, we found $\hat{\delta}(c'_0)$ by minimizing the squared error

$$g(\delta) = \sum_{c_i \in c'_0} \left[ \Phi\left(\Phi^{-1}\left(s_j\left(c'_0 | Y^s, X^s, Y^{MK}\right)\right) + \delta\right) - s_j\left(c_i | Y^s, X^s\right)\right]^2$$

and then defined $s_j(c_i | Y^s, X^s, Y^{MK}) = \Phi\left(\Phi^{-1}(s_j(c'_0 | Y^s, X^s, Y^{MK})) + \hat{\delta}(c_0)\right)$ for $c_i \in c'_0$. Down- and upsampling were only necessary to test the DT in simulations; the daily model was run at a resolution of 1 ha.

## Computational implementation

We implemented the posterior updating pipeline in central processing unit (CPU) partition of the Mahti high-performance computing (HPC) cluster operated by CSC, leveraging parallel processing across multiple species for computational efficiency and scalability. The pipeline is represented by the following set of scripts, which are divided into three distinct stages.

### Data preparation and organization (scripts A0, A1 and A2)

A0_species_dir_arrange.py: This script takes the raw input data, including prior predictions, singing and migration parameters, and prior spatial distribution maps, and organizes them into a structured directory system, creating a dedicated directory for each species. The script preprocesses initial species-specific data and prior information for subsequent steps.

A1_process_meta.py: This script processes the main observation metadata (XData) and migration prior parameters. This adds derived features to the observation data (for example, log_duration and recording type—direct, interval or point) and standardizes species names in the migration parameters. These processed metadata are saved for use in the modelling stage.

A2_prepare_species_data.py: This script is designed to be run per species (indicated by the species_id argument, suitable for a Slurm array job). This reads species-specific spatial prior maps (mean 'a' and variance of the probit model's linear predictor 'vaL') and performs calculations, notably deriving a variance map in the probability space 'va' using a Monte Carlo approximation approach. The resulting spatial variance map is saved for consecutive modelling.

### Species-level model updating and evaluation (script B1)

B1_eval_species.py: This is the core modelling script, run per species (indicated by the species_index argument, suitable for a Slurm array job). This loads the processed metadata (from A1) and species-specific data including spatial priors (from A0 and A2). For each species, training and evaluation is conducted for the following statistical models:

- Detection: updating detection probability based on observation data and recording characteristics.
- Migration: updating migration parameters based on observation data and location/day.
- Spatial distribution: updating spatial distribution probabilities using geographically weighted regression based on observed data and prior spatial maps.

- This script calculates a set of evaluation metrics (AUC, $R^2$, prevalence and likelihood) comparing prior and updated model performance over the specified test period. On request, the updated spatial distribution maps and forecasting results are saved.

## Post-processing and aggregation (scripts C1 and C2)

C1_postprocess.py: This script aggregates the evaluation results from the individual species runs of B1. This reads the evaluation metrics saved by B1 for different model configurations (for example, different training periods or prior types) and compiles them into a single summary table (CSV file). This is used for overall analysis and comparison of model performance.

C2_realtime.py: This script is designed for evaluating the pipeline's performance in a real-time analysis simulation using the retrospective 2024 data. This iterates through sequential training windows, effectively simulating the daily updates shown in the diagram (Fig. 5a). For each time step, the predictions generated by B1 are aggregated with training data over the relevant time frame and evaluation metrics are calculated for a 1-week forecast window. This provides insights into how the model performs with updated parameters as new data are incorporated over time.

In summary, the above-described pipeline prepares data, updates species-specific statistical models in parallel using MK app and prior information (covering detection, migration and spatial components), evaluates the performance of these updated models and then aggregates the results for performance evaluation and downstream visualization. The use of Slurm array jobs allows the computationally intensive modelling step (B1, and data preparation step A2) to be scaled efficiently across many species.

We used the pipeline for three different tasks: (1) modelling of retrospective 2023 and 2024 annual data, (2) simulating real-time daily updates using retrospective 2024 data and (3) conducting real-time daily updates and predictions from the DT throughout the migration season of 2025.

## Evaluation of predictive performance against future MK data

One aim of the DT was to provide real-time updated predictions of future MK app detections. We assessed the quality of these predictions with a walk-forward evaluation on the 2024 data. For computational convenience, we first pre-estimate the posterior detection model using 2023 data. Then, for $T = 0, 1, …, 365$, we trained the migration and spatial models using all data up to time $T$ of the leap year 2024 and predicted over the next day $T + 1$. The predictions were aggregated over the 366 test periods. We evaluated predictive performance with AUC.

## Evaluation of predictive performance using expert point count data

While the above-described evaluation of predictive performance was based on splitting the data temporally into training partition (MK data until present day) and unseen testing partition (MK data for the next day), the test data were not fully independent from the training data. Most importantly, the machine-learning-based classifications may contain consistent mistakes both in the training and in the test data, potentially inflating the apparent predictive performance. To see why this could be the case, assume that MK app detections of species A would be based on misclassification, as in reality originating from species B. Validation against next-day MK app data could yield optimistic results, because both the predictions and the next-day data might agree that A is present, even if this was not the case. Instead, manual point counts by bird experts would correctly suggest that species B is present instead of species A and thus avoid circularity in validation. To compare the performance of prior and posterior models against an independent dataset of bird occurrences, we organized a field campaign involving volunteers (expert birdwatchers) at preselected sites. The campaign consisted of three main steps:

- Volunteer recruitment. We invited birdwatchers to participate via an online survey, which was distributed through local birdwatching mailing lists in Finland. To ensure sufficient birdwatching skills, we evaluated responses on the basis of the volunteers' experience.
- Site selection and assignment. Point count locations were algorithmically selected to maximize contrast between prior and posterior model predictions. Each site included five-point count locations: one central point and four at the corners of a 1-ha square. Volunteers received daily site assignments via Google MyMaps links, each containing 10 sites (that is, 50 daily point count locations). The volunteers could then choose which sites to visit based on accessibility. Volunteers were unaware of the model predictions used to select the sites.
- Fieldwork and data collection. Fieldwork took place from 1 May to 5 June 2025, typically between 5:00 and 10:00, when bird activity is highest. Volunteers recorded species lists (heard or seen) at each point count location, resulting in five lists per site. They submitted the data daily via an online form, and volunteers had the opportunity to review their submissions before analysis.

Sites for point-count locations were assigned to volunteers based on a utility function that prioritizes sites where the posterior and prior differ substantially, while also avoiding oversampling the same scenario across days (for example, repeatedly exploring areas where the posterior is much larger than the prior). As the bird experts conducting manual point counts were unaware of the prior and posterior predictions, the algorithmic selection of the validation sites does not bring bias to the results. The utility was defined in terms of absolute difference between the species' prior spatial distributions and posterior spatial distributions calculated from 2023 and 2024 data and was further modulated by expectation of species provided by prior migration component. Let $c_{<t}$ be the sites selected up to day $t$. For species $j$, we defined the utility $U_j(c_t|c_{<t}) = f_j(c_t) \prod_{i=1}^{t-1}(1 - g_j(c_t, c_i))$, where

$$f_j(c_t) = \left| s_j(c_t|Y^s, X^s, Y^{MK}) - s_j(c_t|Y^s, X^s) \right|$$

and

$$g_j(c_t, c_i) = \gamma M_j(c_i, i) \exp\left( \frac{\tau}{2} \left( \left(s_j(c_t|Y^s, X^s) - s_j(c_i|Y^s, X^s)\right)^2 + \left(s_j(c_t|Y^s, X^s, Y^{MK}) - s_j(c_i|Y^s, X^s, Y^{MK})\right)^2 \right) \right).$$

The first term, $f_j$, promotes sampling of areas where the prior and posterior differ but is agnostic to the direction of the difference and the absolute prior/posterior values. For example, cells with prior/posterior values of $(0.3, 0.8)$, $(0.8, 0.3)$ and $(0.1, 0.6)$ all result in $f_j(c_t) = 0.5$. Selecting sites by naively optimizing only this term would probably provide poor coverage of the different prior/posterior scenarios, resulting in a brittle comparison of the prior and posterior models.

The second term penalizes repeat sampling of prior/posterior scenarios from previous days. The function $g_j$ defines a Gaussian kernel with precision $\tau$ in probability space and is close to 1 (hence, the utility is close to zero) when the prior/posterior values in a candidate cell $c_t$ are close to the prior/posterior values in a previously sampled cell. The migration-based indicator multiplier $M_j(c_i, i) = 1(m_j(c_i, t_i|Y^m) \geq 0.75)$ was introduced to negate the penalizing effect on previously sampled cells where the migrating focal species had not yet arrived. We selected $\gamma = 0.95$ to improve numerical stability by preventing exact zeros and $\tau = 400$, which corresponds to a standard deviation of 0.05 in probability space.

On day $t$, we selected cells to maximize the total migration-modulated utility

$$U(c_t|c_{<t}) = \sum_{j=1}^{p} M_j(c_t, t) U_j(c_t|c_{<t}).$$

Sites were selected sequentially for volunteers on each day subject to hierarchical spatial constraints designed to minimize travel time within days and spatial autocorrelation across days. For volunteer $i$, we first chose a central site

$$c_{t,i,1} = \mathrm{argmax}_c\, U(c|c_{<t})$$

within a 50-km radius of $i$'s central coordinates. We then chose nine other sites sequentially,

$$c_{t,i,k} = \mathrm{argmax}_c\, U(c|c_{<t}),$$

where $k = 2, …, 10$, within a 10-km radius of $c_{i,t,1}$. Volunteers surveyed as many of these sites as possible in a given morning. Sites were chosen to be at least 1 km away from previously selected sites.

At each site, we allocated 5 point-count locations placed in the corners and centre of a $100 \times 100$-m square centred at the site coordinates.

The survey resulted in 1,185 point-count observations conducted at 245 sites on days $T = 126, …, 157$ corresponding to 1 May to 5 June interval of 2025. Following our walk-forward evaluation design, for each day $T$ we extracted the predictions of the migration and spatial posterior models from the DT that was trained with all MK data up to day $T − 1$, as well as predictions of the prior model for detection. We compared the product of these three prediction components against the corresponding species occurrence data collected in the survey, computing the AUC value for each species separately.

We compared the DT predictions with eBird Status and Trends Weekly Abundance Maps geospatial data product, which has weekly temporal resolution. We extracted the species occurrence predictions for the spatial locations of each point-count observation in the week for which the centre was closest to the date of the point count. As the currently available eBird Status and Trends Weekly Abundance Maps geospatial data product covers only 53 species out of 80 that we included in the survey, we replicated the subsequent AUC calculation for these species only.

## FAIR publication of the data in FinBIF

Occurrence records produced by the MK app are copied to FinBIF[65] and published on the Laji.fi portal. The dataset has a persistent identifier (http://tun.fi/HR.6578) and metadata describing, for example, creation method, ownership, licensing, citation and details about the machine-learning-based classifier used. Observations are retrieved from CSC's storage servers. Only records with a species-specific confidence value ≥0.9 are retained. Privacy and species protection measures are applied, including coordinate rounding ($1 \times 1$ km Finnish Uniform Coordinate System grid), pseudonymization and 1–100 km obfuscation for sensitive taxa[66]. These observations are archived and available for download in their original format and full detail.

Observations are aggregated by species, date and location, with summary attributes such as detection count and maximum confidence. The dataset is converted to the FinBIF schema, validated and ingested into the FinBIF data warehouse via an API. It is published as FAIR, open-access data on Laji.fi, where records are flagged as machine observations for filtering. They are searchable using the Laji.fi observation search and are also synchronized to the Global Biodiversity Information Facility[67]. In addition, exact non-obfuscated locations are stored in FinBIF's public-authority portal and are available only to Finnish authorities or through formal data requests[66].

Any registered FinBIF user can flag potentially incorrect records for expert review. Assigned experts can review questionable records based on associated audio, which is not public due to privacy reasons[68].

## Reporting summary

Further information on research design is available in the Nature Portfolio Reporting Summary linked to this article.

## Data availability

The data generated by the MK app and the associated metadata are available via FinBIF at the persistent identifier http://tun.fi/HR.6578. According to eBird platform policy, access to the Status and Trends Weekly Abundance Maps can be obtained through online application request on the platform's website. Interactive examples of DT predictions are available at https://mk-app-realtime.projects.earthengine.app/view/mk-app-realtime-test. Source data are provided with this paper.

## Code availability

The source code of the DT as well as the source code by which the prior predictions were generated are available via Zenodo at https://doi.org/10.5281/zenodo.15774443 (ref. 69).

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

## Acknowledgements

We thank the Finland's national public broadcasting company Yle for great collaboration in running the citizen science campaign, in particular managing editor V. Alijoki, producers O. Koski and K. Ström and journalists M. Pyykkö, A. Hauta-aho, M. Peltola and K. Kotakorpi and emeritus journalist and biologist V. Neuvonen. We thank Metsähallitus, the towns of Helsinki and Jyväskylä, and other collaborators who contributed to the set-up of the point count network. We thank LUMA Centre Finland and BirdLife Finland for help in organizing citizen science events for school children. We thank P. Lehikoinen, S. Andrejeff and N. Paulaniemi for their contributions in annotating bird data for model training and calibration. We thank J. Lundén, J. Södersved, S. Neuvonen, J. Sundström, I. Koskinen and P. Uotila for taking part in the expert model validation. CSC – IT Center for Science Ltd. is acknowledged for providing the computing resources to run the backend services and store the data generated by the MK app. In particular, we thank all the citizens who participated in the project and contributed data. We acknowledge funding from the Research Council of Finland (grant nos. 336212 and 345110 to O.O.); the European Research Council (ERC) under the European Union's Horizon 2020 research and innovation programme (grant no. 856506: ERC-synergy project LIFEPLAN to O.O., D.D. and T.R.; grant no. 101123091: ERC-PoC project 'Breaking the wall between professional science and citizen science by hyperautomation' to O.O.); the HORIZON-INFRA-2021-TECH-01 grant 101057437 (Biodiversity Digital Twin for Advanced Modelling, Simulation and Prediction Capabilities to A. Kallio, J.H., G.Z. and O.O.); the Jane and Aatos Erkko Foundation (grant to establish Digital Citizen Science Centre for 2024–2028 to O.O. and A. Lehtiö), NSF IIS-2426762 (D.D.), and FinBIF FIRI 2021 funded by the Research Council of Finland (M.H.).

## Author contributions

O.O. conceived the idea, led the project, contributed to statistical modelling and wrote the first draft of the manuscript. S.W. contributed to and implemented statistical modelling and contributed to the first draft of the manuscript. G.T. contributed to statistical modelling and implemented high-performance computations. P.L. implemented the classification model and performed expert model validation. A. Lehtiö led the development of the smartphone application. O.O., S.W., G.T., P.L. and A. Lehtiö contributed equally to the work. O.N. coordinated and performed expert model validation and participated in the implementation of the citizen science campaign. N.A. contributed to manuscript preparation. A.A. participated in the implementation of the citizen science campaign, in particular school collaboration. J.P.H. led the BioDT project, contributed to the concept of the DT and coordinated computational backend development. M.H. contributed to the development of data management pipelines. A. Kallio contributed to the development of the computational backend. A. Koliseva participated in the implementation of the citizen science campaign, in particular school collaboration. A. Lehikoinen provided long-term monitoring data and commented on the scientific approach. T.R. contributed to manuscript preparation. P.S. contributed to the classification model. A.T.S. contributed to the implementation of the user portal. J. Tahir contributed to the development of the computational backend. J. Talaskivi contributed to the development of the smartphone application and application programming interface specifications. A.T. contributed to the development of data management pipelines. A.V. contributed to the development of the computational backend. G.Z. led the BioDT project and coordinated computational backend development. Several authors collected PAM data at Kilpisjärvi Biological Station (H.A., O.K., M. Sujala and S.V.), Archipelago Research Institute (J.H. and J.I.), Konnevesi Research Station (J.K., S.T. and P.C.W.), Lammi Biological Station (M.K., J.L., J.S., E.-P.T. and J.U.), Kiiminki Field Site (M.M., M.V. and E.V.), Oulanka Research Station (R.P.), Hyytiälä Forest Station (P.S.-A.) and Värriö Subarctic Research Station (M. Sipilä). K.K., M.O. and R.R. performed expert model validation. D.D. supervised statistical modelling.

## Funding

## Competing interests

The authors declare no competing interests.

## Additional information

**Correspondence and requests for materials** should be addressed to Otso Ovaskainen.

[1]Department of Biological and Environmental Science, University of Jyväskylä, Jyväskylä, Finland. [2]Department of Statistical Science, Duke University, Durham, NC, USA. [3]Organismal and Evolutionary Biology Research Programme, Faculty of Biological and Environmental Sciences, University of Helsinki, Helsinki, Finland. [4]Digital Services, University of Jyväskylä, Jyväskylä, Finland. [5]Open Science Centre, University of Jyväskylä, Jyväskylä, Finland. [6]Faculty of Mathematics and Science, University of Jyväskylä, Jyväskylä, Finland. [7]CSC – IT Center for Science Ltd., Espoo, Finland. [8]The Finnish Museum of Natural History Luomus, University of Helsinki, Helsinki, Finland. [9]Department of Ecology, Swedish University of Agricultural Sciences, Uppsala, Sweden. [10]Ecosystems and Environment Research Programme, Faculty of Biological and Environmental Sciences, University of Helsinki, Helsinki, Finland. [11]Institute for Atmospheric and Earth System Research, Forest Sciences, Faculty of Agriculture, University of Helsinki, Helsinki, Finland. [12]Kilpisjärvi Biological Station, University of Helsinki, Helsinki, Finland. [13]Archipelago Research Institute, Biodiversity Unit, University of Turku, Turku, Finland. [14]Konnevesi Research Station, University of Jyväskylä, Konnevesi, Finland. [15]Lammi Biological Station, Faculty of Biological and Environmental Sciences, University of Helsinki, Lammi, Finland. [16]School of Chemical Engineering, Department of Chemistry and Materials Science, Aalto University, Aalto, Finland. [17]Ecology and Genetics Research Unit, Faculty of Science, University of Oulu, Oulu, Finland. [18]Oulanka Research Station, University of Oulu Infrastructure Platfrom, Kuusamo, Finland. [19]Faculty of Medicine, University of Helsinki, Helsinki, Finland. [20]Faculty of Science, Institute for Atmospheric and Earth System Research/Physics, University of Helsinki, Helsinki, Finland. [21]Zoological Museum, Biodiversity Unit, University of Oulu, Oulu, Finland. [22]These authors contributed equally: Otso Ovaskainen, Steven Winter, Gleb Tikhonov, Patrik Lauha, Ari Lehtiö. ✉e-mail: otso.t.ovaskainen@jyu.fi

# Reporting Summary

## Statistics

For all statistical analyses, confirm that the following items are present in the figure legend, table legend, main text, or Methods section.

| n/a | Confirmed | |
|---|---|---|
| ☐ | ☒ | The exact sample size (*n*) for each experimental group/condition, given as a discrete number and unit of measurement |
| ☐ | ☒ | A statement on whether measurements were taken from distinct samples or whether the same sample was measured repeatedly |
| ☐ | ☒ | The statistical test(s) used AND whether they are one- or two-sided *Only common tests should be described solely by name; describe more complex techniques in the Methods section.* |
| ☐ | ☒ | A description of all covariates tested |
| ☐ | ☒ | A description of any assumptions or corrections, such as tests of normality and adjustment for multiple comparisons |
| ☐ | ☒ | A full description of the statistical parameters including central tendency (e.g. means) or other basic estimates (e.g. regression coefficient) AND variation (e.g. standard deviation) or associated estimates of uncertainty (e.g. confidence intervals) |
| ☐ | ☒ | For null hypothesis testing, the test statistic (e.g. *F*, *t*, *r*) with confidence intervals, effect sizes, degrees of freedom and *P* value noted *Give P values as exact values whenever suitable.* |
| ☐ | ☒ | For Bayesian analysis, information on the choice of priors and Markov chain Monte Carlo settings |
| ☐ | ☒ | For hierarchical and complex designs, identification of the appropriate level for tests and full reporting of outcomes |
| ☒ | ☐ | Estimates of effect sizes (e.g. Cohen's *d*, Pearson's *r*), indicating how they were calculated |

*Our web collection on statistics for biologists contains articles on many of the points above.*

## Software and code

Policy information about availability of computer code

| Data collection | Computer code was not used for data collection |
|---|---|
| Data analysis | The digital twin was implemented using R and python. The source code of the Digital Twin is available at https://doi.org/10.5281/zenodo.15774443. |

For manuscripts utilizing custom algorithms or software that are central to the research but not yet described in published literature, software must be made available to editors and reviewers. We strongly encourage code deposition in a community repository (e.g. GitHub). See the Nature Portfolio guidelines for submitting code & software for further information.

## Data

Policy information about availability of data

All manuscripts must include a data availability statement. This statement should provide the following information, where applicable:
- Accession codes, unique identifiers, or web links for publicly available datasets
- A description of any restrictions on data availability
- For clinical datasets or third party data, please ensure that the statement adheres to our policy

The data generated by the MK app and the associated metadata are available at the Finnish Biodiversity Information Facility (FinBIF) at the persistent identifier http://tun.fi/HR.6578. Examples of daily updating predictions are available at https://mk-app-realtime.projects.earthengine.app/view/mk-app-realtime-test.

# Research involving human participants, their data, or biological material

Policy information about studies with human participants or human data. See also policy information about sex, gender (identity/presentation), and sexual orientation and race, ethnicity and racism.

| | |
|---|---|
| Reporting on sex and gender | Citizen scientists who took part to the research were anonymous so their sex and gender are unknown to us. |
| Reporting on race, ethnicity, or other socially relevant groupings | Citizen scientists who took part to the research were anonymous so their race, ethnicity, or other socially relevant groupings are unknown to us. |
| Population characteristics | Citizen scientists who took part to the research were anonymous so their population characteristics are unknown to us. |
| Recruitment | The MK application was launched in collaboration with Finland's national public broadcasting company Yle, which significantly amplified its visibility in national media. The first public mention occurred on April 12th, 2023, during Metsäradio ("Forest radio" in Finnish) program, which focuses on forestry, nature and outdoor lifestyle. Subsequent coverage included the Luontoilta ("Nature Evening" in Finnish) radio broadcast on May 4th, and a featured theme on YLE's special television program Muuttolintujen Kevät ("Spring of Migratory Birds" in Finnish) on May 10th. Given the large interest the topic received among the general public, the application was also highlighted in Yle's main evening news broadcast, which reaches an average television audience of approximately 750,000 viewers – roughly 14% of the Finnish population. In addition to traditional media, Yle promoted the citizen science campaign through its social media channels throughout the spring. Simultaneously with Yle's campaign, the University of Jyväskylä organized citizen science events for a local audience during three consecutive springs. During these events, citizens had the possibility to interact with scientists about topics related to the MK application, and more broadly about birds and environmental change. The MK app and the citizen science events received attention in several local and national newspapers, as well as in birdwatching-related communities such as local BirdLife partners.

School collaboration is an effective way to inspire children and youth to explore science and engage in citizen science. The MK app was developed to work also in the educational setting and in science education. In spring 2024, we published educational material to help teachers integrate the MK app into their teaching. The material, which supports Finland's national curriculum for basic education, is freely available in Finnish and Swedish (https://mappa.fi/materiaalit/muuttolintujen-kevat-sovellus). This material was developed in collaboration with the Finnish Association of Nature and Environmental Schools and the Central Finland LUMA Centre. The educational use of the MK app was tested nationally in the "Suuri Linturetki" (Great Bird Excursion) event aimed at primary schools, as well as in the LUMA Centre Finland's remote afternoon club for 1st and 2nd graders. Through school collaboration, information about the MK app spread to homes via students, reaching an even larger audience. For many families, the MK app became a hobby that connects generations. The MK application was launched in collaboration with Finland's national public broadcasting company Yle, which significantly amplified its visibility in national media. The first public mention occurred on April 12th, 2023, during Metsäradio ("Forest radio" in Finnish) program, which focuses on forestry, nature and outdoor lifestyle. Subsequent coverage included the Luontoilta ("Nature Evening" in Finnish) radio broadcast on May 4th, and a featured theme on YLE's special television program Muuttolintujen Kevät ("Spring of Migratory Birds" in Finnish) on May 10th. Given the large interest the topic received among the general public, the application was also highlighted in Yle's main evening news broadcast, which reaches an average television audience of approximately 750,000 viewers – roughly 14% of the Finnish population. In addition to traditional media, Yle promoted the citizen science campaign through its social media channels throughout the spring. Simultaneously with Yle's campaign, the University of Jyväskylä organized citizen science events for a local audience during three consecutive springs. During these events, citizens had the possibility to interact with scientists about topics related to the MK application, and more broadly about birds and environmental change. The MK app and the citizen science events received attention in several local and national newspapers, as well as in birdwatching-related communities such as local BirdLife partners. |
| Ethics oversight | The research conducted does not need ethical permit. |

Note that full information on the approval of the study protocol must also be provided in the manuscript.

# Field-specific reporting

Please select the one below that is the best fit for your research. If you are not sure, read the appropriate sections before making your selection.

☐ Life sciences    ☐ Behavioural & social sciences    ☒ Ecological, evolutionary & environmental sciences

For a reference copy of the document with all sections, see nature.com/documents/nr-reporting-summary-flat.pdf

# Life sciences study design

All studies must disclose on these points even when the disclosure is negative.

| | |
|---|---|
| Sample size | *Describe how sample size was determined, detailing any statistical methods used to predetermine sample size OR if no sample-size calculation was performed, describe how sample sizes were chosen and provide a rationale for why these sample sizes are sufficient.* |
| Data exclusions | *Describe any data exclusions. If no data were excluded from the analyses, state so OR if data were excluded, describe the exclusions and the rationale behind them, indicating whether exclusion criteria were pre-established.* |

| Replication | *Describe the measures taken to verify the reproducibility of the experimental findings. If all attempts at replication were successful, confirm this OR if there are any findings that were not replicated or cannot be reproduced, note this and describe why.* |
|---|---|
| Randomization | *Describe how samples/organisms/participants were allocated into experimental groups. If allocation was not random, describe how covariates were controlled OR if this is not relevant to your study, explain why.* |
| Blinding | *Describe whether the investigators were blinded to group allocation during data collection and/or analysis. If blinding was not possible, describe why OR explain why blinding was not relevant to your study.* |

# Behavioural & social sciences study design

All studies must disclose on these points even when the disclosure is negative.

| Study description | *Briefly describe the study type including whether data are quantitative, qualitative, or mixed-methods (e.g. qualitative cross-sectional, quantitative experimental, mixed-methods case study).* |
|---|---|
| Research sample | *State the research sample (e.g. Harvard university undergraduates, villagers in rural India) and provide relevant demographic information (e.g. age, sex) and indicate whether the sample is representative. Provide a rationale for the study sample chosen. For studies involving existing datasets, please describe the dataset and source.* |
| Sampling strategy | *Describe the sampling procedure (e.g. random, snowball, stratified, convenience). Describe the statistical methods that were used to predetermine sample size OR if no sample-size calculation was performed, describe how sample sizes were chosen and provide a rationale for why these sample sizes are sufficient. For qualitative data, please indicate whether data saturation was considered, and what criteria were used to decide that no further sampling was needed.* |
| Data collection | *Provide details about the data collection procedure, including the instruments or devices used to record the data (e.g. pen and paper, computer, eye tracker, video or audio equipment) whether anyone was present besides the participant(s) and the researcher, and whether the researcher was blind to experimental condition and/or the study hypothesis during data collection.* |
| Timing | *Indicate the start and stop dates of data collection. If there is a gap between collection periods, state the dates for each sample cohort.* |
| Data exclusions | *If no data were excluded from the analyses, state so OR if data were excluded, provide the exact number of exclusions and the rationale behind them, indicating whether exclusion criteria were pre-established.* |
| Non-participation | *State how many participants dropped out/declined participation and the reason(s) given OR provide response rate OR state that no participants dropped out/declined participation.* |
| Randomization | *If participants were not allocated into experimental groups, state so OR describe how participants were allocated to groups, and if allocation was not random, describe how covariates were controlled.* |

# Ecological, evolutionary & environmental sciences study design

All studies must disclose on these points even when the disclosure is negative.

| Study description | We created a new smartphone app called "Muuttolintujen Kevät", henceforth the MK app, with the Finnish name meaning "The Spring of Migratory Birds". The app was launched on March 30th, 2023, through a publicity campaign run in collaboration with the Finnish broadcasting company YLE. The MK app was specifically designed to overcome two critical limitations of citizen science in biodiversity research. We developed a Digital Twin (DT) that predicts spatiotemporal distributions of bird occurrences and their vocal activity across Finland, with a spatial resolution of one-hectare and a temporal resolution of one hour. The DT provides both past predictions (January 2023 onward) and future predictions. The DT operates by updating a prior model each night using the latest data accumulated through the MK app. |
|---|---|
| Research sample | Each recording made by a citizen scientist is a separate research sample. One sample may involve vocalizations of one or more bird species. |
| Sampling strategy | Citizens were encouraged to download the app, and then make recordings in three modes: direct recordings, interval recordings, or point count recordings. |
| Data collection | The audio data recorded by citizens were saved in a backend, which classified the data to bird species using a AI-based model. |
| Timing and spatial scale | The data collection started in March 30th 2023 and is still ongoing. The spatial scale of the study is Finland. |
| Data exclusions | The data that the users marked as "test data" were excluded. Statistical analyses were made based on classifications that achieved at least 90% probability of correct classification. |
| Reproducibility | The data collection is not reproducible but all other analyses steps are (most importantly, AI-based classification and statistical analyses). |

| | |
|---|---|
| Randomization | There was no randomizing in the main data collection as the data were collected by citizen scientists. The validation data were collected by bird experts in pre-selected locations that were not randomized but selected to maximize information gain. |
| Blinding | There was no blinding in the main data collection as the data were collected by citizen scientists. The bird experts collecting the validation data were unaware of the prior and posterior predictions for the locations they visited. |

Did the study involve field work? ☒ Yes ☐ No

## Field work, collection and transport

| | |
|---|---|
| Field conditions | Data were collected continuosly over the years and over the times of the day, so the field conditions represented the Finnish conditions. Most data were collected during spring and near human settlement. |
| Location | The data covers entire Finland, being more dense in Southern Finland and near human settlement. |
| Access & import/export | The data generated by the MK app and the associated metadata are available at the Finnish Biodiversity Information Facility (FinBIF) at the persistent identifier http://tun.fi/HR.6578. |
| Disturbance | There was no specific disturbance, as the data collection is based on audio recording. |

# Reporting for specific materials, systems and methods

We require information from authors about some types of materials, experimental systems and methods used in many studies. Here, indicate whether each material, system or method listed is relevant to your study. If you are not sure if a list item applies to your research, read the appropriate section before selecting a response.

## Materials & experimental systems

| n/a | Involved in the study |
|---|---|
| ☒ | ☐ Antibodies |
| ☒ | ☐ Eukaryotic cell lines |
| ☒ | ☐ Palaeontology and archaeology |
| ☒ | ☐ Animals and other organisms |
| ☒ | ☐ Clinical data |
| ☒ | ☐ Dual use research of concern |
| ☒ | ☐ Plants |

## Methods

| n/a | Involved in the study |
|---|---|
| ☒ | ☐ ChIP-seq |
| ☒ | ☐ Flow cytometry |
| ☒ | ☐ MRI-based neuroimaging |

## Antibodies

| | |
|---|---|
| Antibodies used | *Describe all antibodies used in the study; as applicable, provide supplier name, catalog number, clone name, and lot number.* |
| Validation | *Describe the validation of each primary antibody for the species and application, noting any validation statements on the manufacturer's website, relevant citations, antibody profiles in online databases, or data provided in the manuscript.* |

## Eukaryotic cell lines

Policy information about cell lines and Sex and Gender in Research

| | |
|---|---|
| Cell line source(s) | *State the source of each cell line used and the sex of all primary cell lines and cells derived from human participants or vertebrate models.* |
| Authentication | *Describe the authentication procedures for each cell line used OR declare that none of the cell lines used were authenticated.* |
| Mycoplasma contamination | *Confirm that all cell lines tested negative for mycoplasma contamination OR describe the results of the testing for mycoplasma contamination OR declare that the cell lines were not tested for mycoplasma contamination.* |
| Commonly misidentified lines (See ICLAC register) | *Name any commonly misidentified cell lines used in the study and provide a rationale for their use.* |

## Palaeontology and Archaeology

| | |
|---|---|
| Specimen provenance | *Provide provenance information for specimens and describe permits that were obtained for the work (including the name of the* |

| Specimen provenance | *issuing authority, the date of issue, and any identifying information). Permits should encompass collection and, where applicable, export.* |
| Specimen deposition | *Indicate where the specimens have been deposited to permit free access by other researchers.* |
| Dating methods | *If new dates are provided, describe how they were obtained (e.g. collection, storage, sample pretreatment and measurement), where they were obtained (i.e. lab name), the calibration program and the protocol for quality assurance OR state that no new dates are provided.* |

☐ Tick this box to confirm that the raw and calibrated dates are available in the paper or in Supplementary Information.

| Ethics oversight | *Identify the organization(s) that approved or provided guidance on the study protocol, OR state that no ethical approval or guidance was required and explain why not.* |

Note that full information on the approval of the study protocol must also be provided in the manuscript.

# Animals and other research organisms

Policy information about studies involving animals; ARRIVE guidelines recommended for reporting animal research, and Sex and Gender in Research

| Laboratory animals | *For laboratory animals, report species, strain and age OR state that the study did not involve laboratory animals.* |
| Wild animals | *Provide details on animals observed in or captured in the field; report species and age where possible. Describe how animals were caught and transported and what happened to captive animals after the study (if killed, explain why and describe method; if released, say where and when) OR state that the study did not involve wild animals.* |
| Reporting on sex | *Indicate if findings apply to only one sex; describe whether sex was considered in study design, methods used for assigning sex. Provide data disaggregated for sex where this information has been collected in the source data as appropriate; provide overall numbers in this Reporting Summary. Please state if this information has not been collected.  Report sex-based analyses where performed, justify reasons for lack of sex-based analysis.* |
| Field-collected samples | *For laboratory work with field-collected samples, describe all relevant parameters such as housing, maintenance, temperature, photoperiod and end-of-experiment protocol OR state that the study did not involve samples collected from the field.* |
| Ethics oversight | *Identify the organization(s) that approved or provided guidance on the study protocol, OR state that no ethical approval or guidance was required and explain why not.* |

Note that full information on the approval of the study protocol must also be provided in the manuscript.

# Clinical data

Policy information about clinical studies
All manuscripts should comply with the ICMJE guidelines for publication of clinical research and a completed CONSORT checklist must be included with all submissions.

| Clinical trial registration | *Provide the trial registration number from ClinicalTrials.gov or an equivalent agency.* |
| Study protocol | *Note where the full trial protocol can be accessed OR if not available, explain why.* |
| Data collection | *Describe the settings and locales of data collection, noting the time periods of recruitment and data collection.* |
| Outcomes | *Describe how you pre-defined primary and secondary outcome measures and how you assessed these measures.* |

# Dual use research of concern

Policy information about dual use research of concern

## Hazards

Could the accidental, deliberate or reckless misuse of agents or technologies generated in the work, or the application of information presented in the manuscript, pose a threat to:

No | Yes

☐ ☐ Public health

☐ ☐ National security

☐ ☐ Crops and/or livestock

☐ ☐ Ecosystems

☐ ☐ Any other significant area

## Experiments of concern

Does the work involve any of these experiments of concern:

No  Yes

☐  ☐  Demonstrate how to render a vaccine ineffective

☐  ☐  Confer resistance to therapeutically useful antibiotics or antiviral agents

☐  ☐  Enhance the virulence of a pathogen or render a nonpathogen virulent

☐  ☐  Increase transmissibility of a pathogen

☐  ☐  Alter the host range of a pathogen

☐  ☐  Enable evasion of diagnostic/detection modalities

☐  ☐  Enable the weaponization of a biological agent or toxin

☐  ☐  Any other potentially harmful combination of experiments and agents

# Plants

Seed stocks
> *Report on the source of all seed stocks or other plant material used. If applicable, state the seed stock centre and catalogue number. If plant specimens were collected from the field, describe the collection location, date and sampling procedures.*

Novel plant genotypes
> *Describe the methods by which all novel plant genotypes were produced. This includes those generated by transgenic approaches, gene editing, chemical/radiation-based mutagenesis and hybridization. For transgenic lines, describe the transformation method, the number of independent lines analyzed and the generation upon which experiments were performed. For gene-edited lines, describe the editor used, the endogenous sequence targeted for editing, the targeting guide RNA sequence (if applicable) and how the editor was applied.*

Authentication
> *Describe any authentication procedures for each seed stock used or novel genotype generated. Describe any experiments used to assess the effect of a mutation and, where applicable, how potential secondary effects (e.g. second site T-DNA insertions, mosiacism, off-target gene editing) were examined.*

# ChIP-seq

## Data deposition

☐ Confirm that both raw and final processed data have been deposited in a public database such as GEO.

☐ Confirm that you have deposited or provided access to graph files (e.g. BED files) for the called peaks.

Data access links
*May remain private before publication.*
> *For "Initial submission" or "Revised version" documents, provide reviewer access links. For your "Final submission" document, provide a link to the deposited data.*

Files in database submission
> *Provide a list of all files available in the database submission.*

Genome browser session
(e.g. UCSC)
> *Provide a link to an anonymized genome browser session for "Initial submission" and "Revised version" documents only, to enable peer review. Write "no longer applicable" for "Final submission" documents.*

## Methodology

Replicates
> *Describe the experimental replicates, specifying number, type and replicate agreement.*

Sequencing depth
> *Describe the sequencing depth for each experiment, providing the total number of reads, uniquely mapped reads, length of reads and whether they were paired- or single-end.*

Antibodies
> *Describe the antibodies used for the ChIP-seq experiments; as applicable, provide supplier name, catalog number, clone name, and lot number.*

Peak calling parameters
> *Specify the command line program and parameters used for read mapping and peak calling, including the ChIP, control and index files used.*

Data quality
> *Describe the methods used to ensure data quality in full detail, including how many peaks are at FDR 5% and above 5-fold enrichment.*

Software
> *Describe the software used to collect and analyze the ChIP-seq data. For custom code that has been deposited into a community repository, provide accession details.*

# Flow Cytometry

## Plots

Confirm that:

☐ The axis labels state the marker and fluorochrome used (e.g. CD4-FITC).

☐ The axis scales are clearly visible. Include numbers along axes only for bottom left plot of group (a 'group' is an analysis of identical markers).

☐ All plots are contour plots with outliers or pseudocolor plots.

☐ A numerical value for number of cells or percentage (with statistics) is provided.

## Methodology

Sample preparation | *Describe the sample preparation, detailing the biological source of the cells and any tissue processing steps used.*

Instrument | *Identify the instrument used for data collection, specifying make and model number.*

Software | *Describe the software used to collect and analyze the flow cytometry data. For custom code that has been deposited into a community repository, provide accession details.*

Cell population abundance | *Describe the abundance of the relevant cell populations within post-sort fractions, providing details on the purity of the samples and how it was determined.*

Gating strategy | *Describe the gating strategy used for all relevant experiments, specifying the preliminary FSC/SSC gates of the starting cell population, indicating where boundaries between "positive" and "negative" staining cell populations are defined.*

☐ Tick this box to confirm that a figure exemplifying the gating strategy is provided in the Supplementary Information.

# Magnetic resonance imaging

## Experimental design

Design type | *Indicate task or resting state; event-related or block design.*

Design specifications | *Specify the number of blocks, trials or experimental units per session and/or subject, and specify the length of each trial or block (if trials are blocked) and interval between trials.*

Behavioral performance measures | *State number and/or type of variables recorded (e.g. correct button press, response time) and what statistics were used to establish that the subjects were performing the task as expected (e.g. mean, range, and/or standard deviation across subjects).*

## Acquisition

Imaging type(s) | *Specify: functional, structural, diffusion, perfusion.*

Field strength | *Specify in Tesla*

Sequence & imaging parameters | *Specify the pulse sequence type (gradient echo, spin echo, etc.), imaging type (EPI, spiral, etc.), field of view, matrix size, slice thickness, orientation and TE/TR/flip angle.*

Area of acquisition | *State whether a whole brain scan was used OR define the area of acquisition, describing how the region was determined.*

Diffusion MRI        ☐ Used        ☐ Not used

## Preprocessing

Preprocessing software | *Provide detail on software version and revision number and on specific parameters (model/functions, brain extraction, segmentation, smoothing kernel size, etc.).*

Normalization | *If data were normalized/standardized, describe the approach(es): specify linear or non-linear and define image types used for transformation OR indicate that data were not normalized and explain rationale for lack of normalization.*

Normalization template | *Describe the template used for normalization/transformation, specifying subject space or group standardized space (e.g. original Talairach, MNI305, ICBM152) OR indicate that the data were not normalized.*

Noise and artifact removal | *Describe your procedure(s) for artifact and structured noise removal, specifying motion parameters, tissue signals and physiological signals (heart rate, respiration).*

Volume censoring — *Define your software and/or method and criteria for volume censoring, and state the extent of such censoring.*

## Statistical modeling & inference

Model type and settings — *Specify type (mass univariate, multivariate, RSA, predictive, etc.) and describe essential details of the model at the first and second levels (e.g. fixed, random or mixed effects; drift or auto-correlation).*

Effect(s) tested — *Define precise effect in terms of the task or stimulus conditions instead of psychological concepts and indicate whether ANOVA or factorial designs were used.*

Specify type of analysis: ☐ Whole brain   ☐ ROI-based   ☐ Both

Statistic type for inference — *Specify voxel-wise or cluster-wise and report all relevant parameters for cluster-wise methods.*

(See Eklund et al. 2016)

Correction — *Describe the type of correction and how it is obtained for multiple comparisons (e.g. FWE, FDR, permutation or Monte Carlo).*

## Models & analysis

| n/a | Involved in the study |
| --- | --- |
| ☐ | ☐ Functional and/or effective connectivity |
| ☐ | ☐ Graph analysis |
| ☐ | ☐ Multivariate modeling or predictive analysis |

Functional and/or effective connectivity — *Report the measures of dependence used and the model details (e.g. Pearson correlation, partial correlation, mutual information).*

Graph analysis — *Report the dependent variable and connectivity measure, specifying weighted graph or binarized graph, subject- or group-level, and the global and/or node summaries used (e.g. clustering coefficient, efficiency, etc.).*

Multivariate modeling and predictive analysis — *Specify independent variables, features extraction and dimension reduction, model, training and evaluation metrics.*

