## [Peer Review File · Nature Ecology & Evolution]

A Digital Twin for Real-Time Biodiversity Forecasting with Citizen Science Data

Corresponding Author: Professor Otso Ovaskainen

Version 0:

Decision Letter:

29th August 2025

Dear Otso,

Your manuscript entitled "A Digital Twin for Real-Time Biodiversity Forecasting with Citizen Science" has now been seen by three reviewers. You will see from their comments below, they find your work of potential interest but they have raised a number of concerns, some of which major. In light of these comments, we would be interested in considering a major revision along with a response to all the reviewers' points.

We hope you will find the reviewers' comments useful as you decide how to proceed. Please do not hesitate to contact us if there are specific requests from the reviewers that you believe are technically impossible or unlikely to yield a meaningful outcome. However, please bear in mind that we will be reluctant to approach the reviewers again in the absence of appropriate revisions, including additional work to address Reviewer 2's serious criticisms on observer skill evaluation and predictive capacity assessment.

* Provide the manuscript in Microsoft Word format and highlight all changes in the same file or a copy.

* Include a "Response to reviewers" document detailing, point-by-point, how you addressed each referee comment. If no action was taken to address a point, you must provide a compelling argument. This response will be sent back to the referees along with the revised manuscript.

* If you have not done so already we suggest that you begin to revise your manuscript so that it conforms to our Article format instructions at <http://www.nature.com/natecolevol/info/final-submission>. Refer also to any guidelines provided in this letter.

We hope to receive the revisions within 6 months. If you cannot send it within this time, please let us know. We will be happy to consider your revision so long as nothing similar has been accepted for publication at Nature Ecology & Evolution or published elsewhere.

Once ready, please use the link below to submit a revised paper:

Link Redacted

Nature Ecology & Evolution is committed to improving transparency in authorship. As part of our efforts in this direction, we are now requesting that all authors identified as 'corresponding author' on published papers create and link their Open Researcher and Contributor Identifier (ORCID) with their account on the Manuscript Tracking System (MTS), prior to acceptance. This applies to primary research papers only. ORCID helps the scientific community achieve unambiguous

attribution of all scholarly contributions. You can create and link your ORCID from the home page of the MTS by clicking on 'Modify my Springer Nature account'. For more information please visit www.springernature.com/orcid.

[redacted]

Reviewer expertise:

Reviewer #1: digital twins, Bayesian statistics, modelling

Reviewer #2: statistics, participatory science data, bird species monitoring

Reviewer #3: bird population monitoring

Reviewers' comments:

Reviewer #1 (Remarks to the Author):

General

The paper presents a digital twin that models the distribution of more than 200 bird species in Finland in a high spatiotemporal resolution, fed with well-informed priors, expert knowledge, and real-time citizen science data in the form of audio recordings. It accounts for absence/presence as a result of migration, spatial features that predict their presence in suitable habitat, and it controls for detectability of each species given time of the year and day, and sampling method, thereby accounting for sampling biases that are well known to distort citizen-science based ecological research. The key feature of the DT is that it produces relevant and highly up-to-date distribution maps of a large number of avian species, which can be used in environmental management and decision making.

I enjoyed reading the manuscript and I very much support its publication, given that it is a relevant topic with considerable societal impact. I liked the fact that the app used for citizen science data collection reached such a wide audience, which clearly illustrates the societal value and the potential of such DTs for public engagement with nature and wildlife.

Below I added a number of comments that I am not insistent on, but in my view would improve the quality of the manuscript.

Lines 112-115

An important detail is that the app specifically designed with the purpose of the DT in mind. Most CS data collection apps are not. Data are shared and can be used - but in many cases the app designer, citizen user, researcher and modeller have different intentions with the recordings. It would be worth adding a discussion point on this as it has implications for applying this work in other contexts.

Line 148

Here it is not completely clear what is meant by a temporal resolution of one hour. Is that the updating frequency of the DT or are species occurrences modelled on hourly intervals?

Discussion – lines 232-234

I wonder if the authors can comment on the requirements of such a DT that improves predictions of the current and future states of biodiversity – developing an app that needs to reach a wide audience, the IT infrastructure needed to feed the DT with real-time data from the app, access to a supercomputer to re-train the models and calculate the posteriors. How do these aspects weigh against the gains in prediction performance (prior vs posterior) and the requirements to train the prior model? Because intuitively, a change in AUC of 0.06 does not seem like a lot, but I could be wrong here.

And how does the DT performance compare to existing and widely used citizen science platforms for continuous dynamic species distribution modelling such as iNaturalist and eBird? They are mentioned in relation to the remaining challenges with sampling biases, but not in respect to prediction performance of models relying on those data sources. It would be good if the authors can comment on that.

Discussion – lines 267-268

The DT has strong potential in less well studied regions: I agree, this cannot be stressed enough in my view – the added value of DTs (in combination with the Bayesian approach) in data poor cases. Evidence from your research also points to the fact that the approach is particularly powerful for less-studied species. Note that Bayesian approaches have been well-known to enhance species distribution and habitat suitability models in data-poor conditions – see: Hamilton, S. H., Pollino, C. A., & Jakeman, A. J. (2015). Habitat suitability modelling of rare species using Bayesian networks: Model evaluation under limited data. *Ecological modelling*, 299, 64-78.

Discussion – lines 283-284

The approach strengthens the capacity of citizen science to contribute to biodiversity monitoring. But this goes two ways - it's not just citizens contributing to science, but also the fact that such a wide audience is reached with this interactive tool opens

new avenues for public engagement with nature. You have the potential to reach audiences that are less in contact with the natural environment, and you can teach them something about their immediate surroundings. Thereby, DTs (linked with the MK app) can be a powerful new tool to (re)connect contemporary society with nature. I think this can be stressed more - the fact that the app has reached 5% of the Finnish population says a lot about its success!

Figure 2

This reads like a linear modelling process whereby in each DT update the old prior is used as input for the new posterior model. Am I correct to assume that the posterior model is not used in the prior model in the next iteration? Would there be a potential to iteratively update the prior model as well?

Lines 565-567

I wonder to what extent the model is able to correctly identify unusual cases. Or would those cases be lost due to the 90% confidence rule?

Line 613

Is 263 the number of species listed as resident? Or all species ever recorded in Finland? In other words: how do you decide which species? And how to deal with new species previously not recorded?

Lines 622-623 and 627-628

I would remove the phrase 'conditional on ...'. Isn't this condition already implied in the formula - when m_j equals 1 and not 0?

Chapters on prior models

Where can I find the R-scripts used for generating the prior models?

Am I correct to assume that the coefficients of the priors are fixed parameters in the DT?

Chapters on posterior models

I am not sure why the sequence of the chapters is inconsistent with the chapters on the priors.

I must admit that I find this section hard to read and therefore difficult to review. Being able to see the code may help in my understanding but also in replicability of the study.

Computational implementation chapter

I find this chapter very useful. Thank you for this!

Reviewer #2 (Remarks to the Author):

Review for NATECOLEVOL-25072566-T:

"A Digital Twin for Real-Time Biodiversity Forecasting with Citizen Science"

Reviewer Background: PhD-Level applied statistician with experience in statistical learning and causal machine learning methods for population monitoring data. Active areas of research include statistical learning models for animal movement and epidemiological applications for wildlife populations, causal machine learning methodology for trend estimation from structured and opportunistic survey data.

Guiding Questions from the Nature Ecology and Evolution Editorial Board:

Does the manuscript have flaws which should prohibit its publication?

Yes

If the conclusions are not original, it would be very helpful if you could provide relevant references.

Conclusions are original but lack context to support readership understanding.

On a more subjective note, do you feel that the results presented are of immediate interest to many people in your own research area, or to people from several areas?

Yes, but in its current form, I do not believe that reception would be positive or helpful to readership.

If you recommend publication, please outline, in a paragraph or so, what you consider to be the outstanding features.

I do not recommend this work for publication in its current format.

Overall Comments:

The authors present a new data collection app (referred to as the MK app) and methodology (referred to as Digital Twinning [DT]) to predict and forecast species distribution.

Overall, I have several issues with this work that I feel need to be addressed. Terminology is poorly defined and utilized throughout the manuscript. I am not convinced that a general audience at *Nature Ecology and Evolution* could understand the arguments made and presentation of methodology based on the current presentation. I believe this stems from the author's not situating this work with the extensive literature on data integration and species distribution modeling and forecasting. I did not find the presentation of the methodology accommodating of a general quantitative audience that would want to understand modeling decisions made. These issues I have explained in greater detail in the following sections.

Pertaining to the results and discussion, one major concern I have is the lack of commentary on statistical or model-based inference from their model. Environmental variables were used to help estimate species distributions, and the relationships between current environmental conditions and predictions of future species distributions are important. The authors did not provide enough information about the model to help me understand if there is potential for inference about relationships between environmental conditions and future states of a population. As an example, if forecasted distributions decrease in size or detection probability year-after-year can this model help explain potential drivers of this change? If this inference is not easily attainable, it would be a bit of a disappointment and worthy of a disclaimer. I don't think this would be a deal breaker for using methodology such as this, but it would more accurately avoid portraying this work as a panacea.

Predictive performance is a reliable way to assess models such as this, but I think the authors should make more effort to explain the qualitative challenges of using information from this methodology for policy development. I think the relationship between model interpretability and potential to inform policy is frequently overlooked.

Further, I was a bit disappointed to find misleading comparison of the MK app to the eBird project, which is arguably one of the largest and most successful intercontinental citizen science projects in human history in several aspects. This is both from a data perspective (i.e., data volume, data quality, quality control, quantification of user skills through checklist calibration indices) and methodology for citizen science data (i.e., accounting for confounding and changes in the observation process, rigorous statistical simulations, external validation with other structured survey data, etc.). There are some statements made in the discussion that I personally felt were incorrect or had overstatements of the MK app relative to the eBird project. It would be impossible for the MK app to accomplish what eBird has become over the past two decades in a matter of 2-3 years. My advice to the authors would be to limit mentions of eBird or to better familiarize themselves with what has been accomplished with the eBird project in order to more appropriately contextualize the new work with the MK app. The MK app seems like an exciting national-level citizen science project, which I would be excited to see grow further.

Summary/Abstract:

2nd Sentence: Are the issues related to data quality or variance in data quality? As a statistician who develops machine learning models for eBird project data and other more structured surveys (e.g., North American Breeding Bird survey, Saltmarsh Habitat and Avian Research Program, etc.), I am suspicious that the issues is likely variance in data quality. Within the eBird project, there is an exceptional amount of high quality data, but there is also a lot of lower quality data mixed in. I cannot speak for this citizen science platform, but generally for eBird, variance in data quality is the main issue that we seek to address in any modeling problem.

Remaining sentences: I am not 100% sure who the audience is based on this summary in the final sentences. Digital Twinning is a term that I suspect is more familiar to engineers and computer scientists. Coming into reading this paper, I have no idea what this term meant, I have read the definition in the introduction several times and to be honest I didn't understand it until I read the whole paper closely multiple times. I think what makes this confusing is that the fields of conservation ecology, wildlife management, and environmental statistics use the term "data integration" for statistical and/or machine learning modeling uses various sources of data to improve inference. See the following references for some examples:

Zipkin, E. F., Zylstra, E. R., Wright, A. D., Saunders, S. P., Finley, A. O., Dietze, M. C., ... & Tingley, M. W. (2021). Addressing data integration challenges to link ecological processes across scales. *Frontiers in Ecology and the Environment*, 19(1), 30-38.

Zipkin, E. F., Inouye, B. D., & Beissinger, S. R. (2019). Innovations in data integration for modeling populations. *Ecology*, 100(6), 1-3.

I have more comments on this in the introduction.

Main:

-In the second paragraph, it may not hurt to have this citation? I'll leave the authors to decide if citing this manuscript is needed here. A large aspect of the potential of citizen science data research in the U.S. is rooted in semi-structured protocols and the documentation of various aspects of the observation process associated with each observation.

Steve Kelling, Alison Johnston, Aletta Bonn, Daniel Fink, Viviana Ruiz-Gutierrez, Rick Bonney, Miguel Fernandez, Wesley M Hochachka, Romain Julliard, Roland Kraemer, Robert Guralnick, Using Semistructured Surveys to Improve Citizen Science Data for Monitoring Biodiversity, *BioScience*, Volume 69, Issue 3, March 2019, Pages 170–179, <https://doi.org/10.1093/biosci/biz010>

–“Digital Twinning (DT), i.e., the pairing between real-world objects and their digital representations, has been much used in technology to speed up development cycles and to make them more cost-efficient²⁶. While originally developed to simulate engineered systems, there is widespread interest in applying DT to other fields, including biodiversity research. By combining data, models and domain knowledge in a close to real-time alignment with the real world, DT holds considerable potential in ecological forecasting and in enabling rapid environmental decision-making^{27,28}...”

With no offense intended, these 3 sentences mean virtually nothing to me. As a reader, I don't know anything about digital twinning, I don't understand how the term “real world objects” connects to citizen science and I don't know what a “digital representation” of a real world object is. My impression reading this is that domain-experts on this topic carelessly dumped these technical terms into an introduction for an ecology readership. After reading the whole manuscript carefully multiple times and rereading these sentences, I am pretty sure I know what the authors mean here, but requiring this level of effort is not considerate of readership. For a *Nature Ecology and Evolution* manuscript, I suspect the editorial board will appreciate the author's more carefully considering the audience, the language readers are familiar with, and how the work is closely related data integration work of others. How does digital twinning differ from data integration? Are they the same topic with different names??

Now to the next sentences:

–“... However, the development of digital twins for biodiversity remains a complex and emerging research frontier, hindered by the complexity of natural ecosystems, the need to combine heterogeneous data sources and the technical challenges associated with generating and processing real-time biodiversity data streams.”

Combining heterogeneous data sources would be considered data integration to almost any researcher in the Ecological Society of American (ESA) or the British Ecological Society (BES). If you choose to use a different term for the exact same concept, you risk confusing readership. If you choose to use a different term for a slightly different concept, the differences need to be clearly explained somewhere.

Also, what is a “data stream?” Do all ecologists know what a data stream is? What is the difference between a data stream and a data collected from a designed study, opportunistic data from citizen science projects, or data from audio recording units? I know what a data stream is because I have collaborated with computer scientists and data scientists, but I am not sure other readers will immediately know what you are talking about here.

–“This paper aims to demonstrate the applicability of DT approaches in biodiversity research for achieving accurate, real-time predictions of species distributions.”

Integrated species distribution modeling literature seems to be disregarded in this paper:

Ahmad Suhaimi, S. S., Blair, G. S., & Jarvis, S. G. (2021). Integrated species distribution models: A comparison of approaches under different data quality scenarios. *Diversity and Distributions*, 27(6), 1066-1075.

Koshkina, V., Wang, Y., Gordon, A., Dorazio, R. M., White, M., & Stone, L. (2017). Integrated species distribution models: combining presence-background data and site-occupancy data with imperfect detection. *Methods in Ecology and Evolution*, 8(4), 420-430.

Dovers, E., Popovic, G. C., & Warton, D. I. (2024). A fast method for fitting integrated species distribution models. *Methods in Ecology and Evolution*, 15(1), 191-203.

Forester, B. R., DeChaine, E. G., & Bunn, A. G. (2013). Integrating ensemble species distribution modelling and statistical phylogeography to inform projections of climate change impacts on species distributions. *Diversity and Distributions*, 19(12), 1480-1495.

Buisson, L., Thuiller, W., Casajus, N., Lek, S., & Grenouillet, G. (2010). Uncertainty in ensemble forecasting of species distribution. *Global Change Biology*, 16(4), 1145-1157

Lawler, J. J., Wiersma, Y. F., & Huettmann, F. (2010). Using species distribution models for conservation planning and ecological forecasting. In *Predictive species and habitat modeling in landscape ecology: Concepts and applications* (pp. 271-290). New York, NY: Springer New York.

This leads to a major comment. The author's seem to have disregarded literature in ecology on this subject and have made no clear effort in the introduction to connect/situate their work relative to existing and ongoing research. I am an advocate for bringing new methods and ideas to the table in an interdisciplinary setting. However, this paper so far does a disservice to the reader by not helping contextualize the proposed methodology more broadly.

-“This paper aims to demonstrate the applicability of DT approaches in biodiversity research for achieving accurate, real-time predictions of species distributions. We illustrate this through a case study in audio-based bird monitoring, showcasing how reliable real-time biodiversity predictions can be achieved through a DT approach that combines the strengths of citizen science, machine learning, and high-performance computation. Our approach features a continuous model updating process, ensuring that predictions remain responsive to real-time changes in bird activity and environmental conditions. The technological innovations developed in this study not only reduce the time required to generate accurate biodiversity information for policy and management, but also increase inclusivity by broadening the stakeholder community and the roles of the stakeholders. This approach empowers and engages citizens to provide pivotal contributions to both scientific research and environmental monitoring.”

I have mixed feelings about this paragraph. It has a lot of good information that pulls me in as a reader. However, I am left more confused by this paragraph than before I started. The underlined section is where the confusion begins for me.

In this paragraph, there is a first mention of “a case study in audio-based bird monitoring.” Audio-based bird monitoring is often NOT citizen science data. Is the citizen science data different than the audio data? Are the same thing? If there are two sources of citizen science and/or non-citizen science then what are they? Are we then integrating audio data with another data source? HELP! I am lost.

Even if the authors clarify later on, this lack of clarity is not appropriate for a Nature Ecology and Evolution manuscript. Any and all data sources that will be integrated within the DT framework should be clearly listed here to help the reader understand what data sources the authors use to estimate and forecast species distributions.

A new tool for digital citizen science:

-finetuned should be written as fine-tuned

-“First, to mitigate the differences in species identification skills among citizens, all classifications are performed by a machine learning model, with users given the option to confirm or reject the classification.”

To be honest, I find this measure of observer skill to be unsettling. With any species identification application, the quality of the recording, the species family, and environmental conditions can have substantial consequences on classification performance. Observer skill should be assessed relative to other observers. Also what does “reject the classification” have to do with observer skill? Did they reject because they didn’t see the species (and they want to see it to add it to a “life list”)? Did they reject because they know the application is not classifying the species correctly? Did they reject because they did not have the ability in that one instance to follow-up with a confirmed visual of the individual? Did they reject because another individual in their party used playback calls, which were subsequently picked up on a audio recording.

I would hope that other reviewers have a second opinion on this topic, but I do not think basing observer skills off a sound ID app is appropriate. There are other groups working on metrics for observer skills. An example is the checklist calibration index, developed by a group of researchers at Cornell Lab of Ornithology, which measures the relative skill of an observer skill at identifying/finding x number of species for t minutes of search effort and d meters traveled.

Without more detail about how observer skill was measured under this approach, I cannot recommend this work for publication. How observer skill is measured is arguably one of the most important aspects of any citizen science research, and the researchers have made a controversial choice (which is not inherently bad) and then avoided justifying or explaining how this choice is ok.

-“Second, to mitigate spatial observation bias and preferential sampling, the MK app enables not only direct recordings, but also interval recordings and systematic point counts.”

In my opinion, this is extremely poor writing and completely glosses over an introduction of a second data source!

I think this section needs to be rewritten. If there are three data sources, the three data sources should leap off the page and scream at the reader. A numbered list of the 3 data sources would be ideal.

Also, after reading this paragraph, I don’t understand how interval recording mitigates preference sampling or observation bias? Couldn’t observation bias and preferential sampling be made even worse? I don’t think anyone is leaving a phone recording in a park or random location... I would think that almost every sensible adult would participate in interval recordings near home or some secure location. I am suspicious that the “mitigation of bias and preferential sampling” refers to the systematic point counts.

With regards to mitigating preferential sampling and observation bias, I cannot say I am convinced. I would strongly suggest fitting distribution models with each source independently before any type of “twinning” or integration... (still not 100% what the authors are doing exactly). From what I have read so far, I’d be much more conservative with regards to any claims of mitigating bias and issues with preferential sampling unless they can show that the integrated/twinned estimates are improved over disjoint estimation of species distributions. This could be done through simulation but it could also be done through external validation with historical estimates of abundance.

A real-time biodiversity Digital Twin:

-“We developed a Digital Twin (DT) that predicts spatiotemporal distributions of bird occurrences and their vocal activity across Finland, with a spatial resolution of one-hectare and a temporal resolution of one hour.”

I am so sorry. I still don’t know what a digital twin is and how you develop one. (Maybe I’m incompetent and if so I apologize for being assigned as your reviewer)... After multiple hours, here is my guess:

Is DT literally a predictive machine learning model that is fit to multiple data sources to estimate species distributions as a probability of occurrence? If this is true, why not say this? Ecologist should understand this language as it is in-line with ecological statistics literature and standard statistical learning literature.

-“We quantified prior knowledge on bird species’ spatial distributions by fitting the joint species distribution model HMSC31

to long-term data on transect-line surveys, using as predictors one-hectare resolution raster maps of land-use variables, forest structure variables, and climatic variables (Fig. 2B)."

HMSC??? Is that an acronym for some sort of 'Hidden Markov State-Space Model' Perhaps I have missed acronym from earlier?

I am still very confused about the Digital Twin. Is the twin the joint species distribution model? Is it something else?

Beyond this, perhaps the authors were discouraged from including any equations in this work, but I am at a loss for how to evaluate this model. I keep trying to sort through various what variables are being fit to what portions of the model and what model fitting challenges may arise. It may all be valid, but I don't have the time to figure it out as a reviewer, which I apologize for. I can't say that Figure 2 helps me as much as I had hoped when I saw the reference.

Generally speaking, I have some concerns about estimation of species distributions during any migratory period. It makes me extremely uncomfortable. Some species are vocal during migration and some are not... I am not an ecologist so I don't fully understand the utility of real-time forecasting.

Also, when I start to think of species that are detected visually because they are primarily non-vocal (which there are many), I also start to feel less comfortable with what is being presented here.

I am happy to give another go at understanding what the authors are presenting here, but I do not have time to sort through this in it's current state, especially with no frame of reference regarding the similarity or lack thereof between DT and data integration.

Also, I keep getting "MK app data" mixed up with "MK app recording." Are the transect-line surveys recorded within the MK app data? Or are they something different.

I am starting to wondering if it would be good to have a glossary table of definitions and terms.

Example predictions:

No comments for now.

Evaluation of predictive capacity:

-Why is next day performance the metric for assessing predictive capacity? What about next week performance? Next month or next year performance? Is it because the species are not in the MK study area 1 week to 1 month out? My opinion on this should not be the authority, but I was hoping to see longer term forecasting performance assessments.

-"While the above-described evaluation of predictive performance was based on splitting the data temporally into training (MK data until present day) and testing (MK data for the next day), the test data were not fully independent from the training data for two reasons. First, the same citizens may have recorded the same bird individuals on consecutive days. Second, the machine learning based classifications may contain consistent mistakes both in the training and in the test data, potentially inflating the apparent predictive performance."

I do not understand why counting the same bird on consecutive days matters for performance assessment. For resident birds, this is going to happen and it is a critical characteristic of species distributions. If this is really important, I think the authors should clearly explain why this is an issue.

You have lost me again... What do you mean by "the machine learning based classification may contain mistakes." This sentence is utterly useless because all machine learning models contain classification mistakes. If they don't then they are almost certainly overfitting to the data. Are these misclassifications of species based on audio recordings? Or are you referring to the machine learning model that you are using to fit a model to the trained data? I suspect that this second sentence is going to confuse people more than it will help.

-"To evaluate the difference between the posterior and prior predictions against fully independent data, we further performed manual point counts by bird experts in pre-selected locations from May 7st to June 7th, 2025."

What is a "bird expert?" Is this a ornithologist? An seasoned volunteer bird-watcher?

-"The manual point count locations were selected algorithmically to include sites where the prior and posterior predictions were as contradictory as possible."

If this is a form of external validation (using completely independent data), then why does this choice matter? Doing external validation with strong algorithmically-assigned preferential sampling sounds odd to me. This should be justified.

Discussion:

2nd paragraph- I am not a fan of this paragraph, particularly the manner in which eBird is juxtaposed to the MK app.

-"However, the use of these massive citizen science data resources for scientific inference has remained challenging, in particular due to data quality issues such as sampling biases and detection errors²³. For example, the best practice recommendations for using eBird data involve choices related to filtering the data for complete checklists, performing spatial subsampling, and using filters for observation effort³⁶."

Even structure surveys such as the North American Breeding Bird Surveys has sampling biases (e.g., roadside surveys have road-side survey bias) and detection errors. Human participants or remote detection equipment will inherently fail to detect some species.

Complete checklists in eBird have all information regarding location, time, day or year, effort distance, effort hours, number of observers, stationary vs. traveling, and although it is not currently used in most general applications, the eBird project contain GPS tracks for a large majority of checklists which provides important information about how far search effort disperses from the recorded location of the checklist. The eBird project data is completely independent of the Merlin Sound ID app (until recently). The eBird project also contains data from more structured survey efforts such as breeding bird atlas projects.

With eBird data, observer effort can be quantified through a "checklist calibration index." There are several papers regarding method development for eBird data that utilize this index.

-"A core feature of the MK app is that it was directly developed to overcome the outstanding challenges of citizen science²³.

First, to tackle the issue of variable and often unknown sampling effort, the MK app quantifies the location, time, type, and duration of each recording, and implements standardized interval recordings and permanent point count routes. Second, to remove observer heterogeneity in species identification, the MK app uses machine learning-based classifications with well-calibrated estimates of uncertainty. These characteristics of the MK app data facilitated their straightforward integration into a

predictive DT approach without the need for statistically controlling for sampling artefacts.”

-artefacts ... should be artifacts?

-The first and second sentence of this quote makes me wonder if the authors have read the Kelling et al. 2019 paper mentioned previously? The authors are selling the MK app as if it has accomplished something (regarding overcoming the challenges of citizen science) that eBird hasn't. I don't think it is true. Reading this work does not convince me that this statement is appropriate as well.

-The third sentence is even more concerning. The authors have NOT clearly explain how the MK app uses machine learning based classifications to remove observer heterogeneity in species identification. This is a big and controversial idea (especially if it relates to using audio recordings to judge the skill level based on what users end up reporting for a list of species during a given search effort). The authors provide no justification or evidence that this is appropriate and I could not find any mention of appendices where this choice was rigorously evaluated.

-I can understand why the comparison to eBird could be discussed in this paper, but this comparison really falls short. I think part of it is caused by the fact that the MK app is newer and records observations directly through audio recording and through other forms. The eBird app does not directly allow for audio files to be converted into checklists. There is a newer way to do that from the Merlin Sound ID application but this does not represent a large portion of the data collected over the past +20 years.

See also my major comment in the Overall Comments section about interpretability and inference.

Reviewer #2 (Remarks on code availability):

The paper needs to be more clearly written and contextualized with existing literature before any further work by a reviewer.

Reviewer #3 (Remarks to the Author):

This manuscript presents a system (digital twin) for using data collected by citizen scientists to predict real time changes in species distributions. Using an app developed by the authors citizen scientists are able to submit their data directly to the system which then uses this data to update models the following day. The authors show that this method greatly improved predictions of species distributions especially for migrants.

I really like this paper, the method and application are exciting and novel and very clearly explained throughout. The engagement by the public in this app is fantastic and as outlined it can greatly help educate people about the bird species in their environment while also collected extremely useful data. By using several data collection methods the eliminate much of the bias associated with collecting such data and this seems like an important step forward.

I can not find any flaws in this manuscript which would prevent its publication, and I find that the system/method and results presented would be very interesting to people in this field. Although I am not able to follow all the equations and algebra within the methods section, I find the logic presented behind how the model is structured and implemented sound.

I do not have any suggestions for improvements for this work (which is very unlike me).

Catriona Morrison

Version 1:

Decision Letter:

14th November 2025

Dear Dr. Ovaskainen,

Thank you for submitting your revised manuscript "A Digital Twin for Real-Time Biodiversity Forecasting with Citizen Science" (NATECOLEVOL-25072566A). On the basis of the reviewers' comments (reports below), we will be happy in principle to publish it in Nature Ecology & Evolution, pending minor revisions to satisfy the reviewers' final requests and to comply with our editorial and formatting guidelines.

[redacted]

Reviewer #1 (Remarks to the Author):

Thank you for responding to and addressing my comments. I have no further comments.

Reviewer #2 (Remarks to the Author):

The author's have address most of my concerns for this manuscript. I have a few more:

-I would recommend removing the word "merely" in front of data integration, which strongly understates the methodological work in model-based data integration, which in many cases focuses on statistical inference and parameter interpretation. I think there is better language to distinguish data integration literature from DT methods.

-"By relying solely on machine learning based bird classifications rather than citizen-based classifications, we remove an important part of observer heterogeneity and increase inclusivity by enabling ordinary citizens without bird identification skills to take part in data collection by making bird recordings."

I am not a fan of this sentence. Machine learning based bird classifications (generally speaking) are wrought with potential challenges (and biases) in practice. Recording applications such as Merlin filter classification to only "possible birds" based on historical rarity of occurrence. Machine learning based classifications are also prone to the same types of misclassification that occur by human ears (e.g., imitation calls of mockingbirds, blue jays imitating a red-shouldered hawk, etc.). Weather conditions make ML classification even more challenging, and experienced participants will often correctly identify birds that a recording using ML will not detect. Further, I am not convinced that machine learning based classification increases inclusivity. Recording applications (e.g., Merlin) have increased inclusivity encouraging more people to engage in bird watching, but I think it is odd to claim that machine learning classifications increases inclusivity. Further, (in general) for participants who use recording applications designed to help with identification, it is not necessarily clear whether or not recorded observer classifications relied on help from a ML-based recording application.

In this comment, I am not claiming that machine learning based classification as opposed to participant classification is worse or better. This statement understates the potential quality of participant based classification and how it can in some cases be better than and ML based classification. If I were the authors, I would downplay this choice as much as possible unless they have evidence that, in their application, ML based classification improved ____ performance in some way over participant based classification.

Reviewer #1 (Remarks to the Author):

General

The paper presents a digital twin that models the distribution of more than 200 bird species in Finland in a high spatiotemporal resolution, fed with well-informed priors, expert knowledge, and real-time citizen science data in the form of audio recordings. It accounts for absence/presence as a result of migration, spatial features that predict their presence in suitable habitat, and it controls for detectability of each species given time of the year and day, and sampling method, thereby accounting for sampling biases that are well known to distort citizen-science based ecological research. The key feature of the DT is that it produces relevant and highly up-to-date distribution maps of a large number of avian species, which can be used in environmental management and decision making. I enjoyed reading the manuscript and I very much support its publication, given that it is a relevant topic with considerable societal impact. I liked the fact that the app used for citizen science data collection reached such a wide audience, which clearly illustrates the societal value and the potential of such DTs for public engagement with nature and wildlife. Below I added a number of comments that I am not insistent on, but in my view would improve the quality of the manuscript.

We thank you very much for the positive feedback. As detailed in our responses below, we have taken full advantage of the valuable suggestions.

Lines 112-115

An important detail is that the app is specifically designed with the purpose of the DT in mind. Most CS data collection apps are not. Data are shared and can be used - but in many cases the app designer, citizen user, researcher and modeller have different intentions with the recordings. It would be worth adding a discussion point on this as it has implications for applying this work in other contexts.

We fully agree and have now added these sentences to the discussion: "Our DT approach may not generalize straightforwardly to many existing citizen science data streams, as the seamless integration between the data collection and the real-time predictive modelling was enabled by the fact that the MK app was specifically designed to serve this purpose. While building an operational DT such as the one presented here may initially require more effort than most other citizen science platforms, its capabilities go well beyond what static systems can achieve, as it provides a dynamic approach for forecasting biodiversity"

Line 148

Here it is not completely clear what is meant by a temporal resolution of one hour. Is that the updating frequency of the DT or are species occurrences modelled on hourly intervals?

We have clarified that the updating frequency is 24 hours: "...with a spatial resolution of one-hectare, and a temporal updating frequency of one day". By the hourly resolution we referred to how we model singing activity and hence detection probability, but thanks to the reviewer's comment we realized that the text was misleading and needed to be changed.

Discussion – lines 232-234

I wonder if the authors can comment on the requirements of such a DT that improves predictions of the current and future states of biodiversity – developing an app that needs to

reach a wide audience, the IT infrastructure needed to feed the DT with real-time data from the app, access to a supercomputer to re-train the models and calculate the posteriors. How do these aspects weigh against the gains in prediction performance (prior vs posterior) and the requirements to train the prior model? Because intuitively, a change in AUC of 0.06 does not seem like a lot, but I could be wrong here. And how does the DT performance compare to existing and widely used citizen science platforms for continuous dynamic species distribution modelling such as iNaturalist and eBird? They are mentioned in relation to the remaining challenges with sampling biases, but not in respect to prediction performance of models relying on those data sources. It would be good if the authors can comment on that.

We thank you for the valuable suggestions regarding predictive performance comparison. We have now added an explicit comparison with the dynamic species distributions provided by the eBird platform (Status and Trends Weekly Abundance Maps geospatial data product). We do not compare against the iNaturalist, as they do not publicly provide any temporal resolution for their time-invariant species distribution product, though their product is continuously updated on a monthly basis as new data flows into the platform. We report these new results as follows:

“We further compared the DT predictions to those based on the eBird³⁹ global citizen science project. We extracted for each survey week species occurrence probabilities from eBird Status and Trends Weekly Abundance Maps released in summer 2025, representing data that have accumulated until 2023⁴⁰. For those 53 species for which eBird-based predictions were available, mean AUC was 0.62 for eBird-based predictions and 0.67 for DT predictions.”

Concerning the cost-benefit analysis, we have extended the discussion as follows:

“Compared to this effort, the improvement in predictive power that we reported here may appear moderate: the AUC improved from 0.62 in our prior model to 0.67 in the DT. However, we argue that this improvement is substantial, as the AUC value increased by 42% if compared to the baseline value of 0.50. Instead, the low AUC values are explained by the fact that the predictive task that we targeted is highly challenging. Namely, our test data concerns variation in species detections over a small geographic area (where all the species generally occur) and over a short period (during which all the species were generally present), making it highly challenging to predict in which samples the species were present and in which they were absent.”

Discussion – lines 267-268

The DT has strong potential in less well studied regions: I agree, this cannot be stressed enough in my view – the added value of DTs (in combination with the Bayesian approach) in data poor cases. Evidence from your research also points to the fact that the approach is particularly powerful for less-studied species. Note that Bayesian approaches have been well known to enhance species distribution and habitat suitability models in data-poor conditions – see: Hamilton, S. H., Pollino, C. A., & Jakeman, A. J. (2015). Habitat suitability modelling of rare species using Bayesian networks: Model evaluation under limited data. *Ecological modelling*, 299, 64-78.

We agree and thank you for providing the reference. Nonetheless, we decided not to cite it as its main focus (comparison and combination of data- and expert-based models) is not directly linked to our work and as we were constrained with the length restrictions.

Discussion – lines 283-284

The approach strengthens the capacity of citizen science to contribute to biodiversity monitoring. But this goes two ways - it's not just citizens contributing to science, but also the fact that such a wide audience is reached with this interactive tool opens new avenues for public engagement with nature. You have the potential to reach audiences that are less in contact with the natural environment, and you can teach them something about their immediate surroundings. Thereby, DTs (linked with the MK app) can be a powerful new tool to (re)connect contemporary society with nature. I think this can be stressed more - the fact that the app has reached 5% of the Finnish population says a lot about its success!

We thank you for the positive feedback. We have extended the discussion as follows:

“The MK app has substantially promoted citizen engagement and helped reconnect citizens with nature through extensive school collaboration, media coverage, the possibility of sharing results through social media, and educational features such as the bird game. In particular, the MK app has gained popularity among ordinary citizens who do not necessarily recognize any bird sounds themselves, as it enables them not only to learn which birds vocalize in their surroundings, but also to contribute valuable biodiversity data that is immediately used for research and monitoring.”

Figure 2

This reads like a linear modelling process whereby in each DT update the old prior is used as input for the new posterior model. Am I correct to assume that the posterior model is not used in the prior model in the next iteration? Would there be a potential to iteratively update the prior model as well?

This is an excellent question. As you suggest, there is conceptually the potential to update the prior parameters iteratively as new data come in. However, we instead infer the priors using long term data and only update the priors occasionally for computational tractability – which is a critical consideration given the vast quantities of data being used to update our predictions and the (necessary) complexity of our prior models. We now clarify this in the methods section for posterior model for detection: “Since all MK observations are informative about the detection model, after the first year the posterior variance for estimated parameters is already vanishingly small and refitting the model daily with streaming data does not change estimates or performance to any relevant degree. Consequently, we chose to update the detection model once at the start of each year with all available data.”

Lines 565-567

I wonder to what extent the model is able to correctly identify unusual cases. Or would those cases be lost due to the 90% confidence rule?

The 90% confidence rule is applied to exclude large numbers of false positives (where a species is apparently detected but not truly there), which would unavoidably appear unless the noisy data were filtered. For example, vocalizations of a species A may sometimes yield low confidence classifications of another species B that has similar vocalization patterns. Here we refer specifically to cases where there is additional information to judge whether a detection is likely to be a false positive. For example, we may obtain (based on audio only) a detection of a

bird species that is very unlikely to be at that location at that time. For example, the detection may be obtained during winter, even though the species is strictly migratory. In such cases, we downgrade the confidence of the species classification. This is done before the DT modelling, as we wish to apply this filtering step already to the classifications that the user sees in the app. In theory, the modeling approach in the DT could also filter out such observations, as for those data points the prior could dominate. Nonetheless, we found it safer to filter out highly unlikely observations from the beginning. To clarify that this concerns only obvious misclassifications, we have modified the text as “The predictions of highly unlikely species are penalized based on location and day of the year to remove obvious misclassifications (e.g., migratory species detected during winter) from the data.”

Line 613

Is 263 the number of species listed as resident? Or all species ever recorded in Finland? In other words: how do you decide which species? And how to deal with new species previously not recorded?

The 263 bird species considered by our model equal the full breeding fauna as well as all regular vagrants. The model does not cover some 233 species occasionally recorded as stray individuals in Finland, nor does it cover species not previously recorded – since both sets represent extremely unlikely detections. We have now clarified this issue in the main text as follows: “...263 Finnish bird species (all breeding species, non-breeding migrants and most common vagrants)”. In the Methods section, we further write “While the list of the selected 263 species is not the full list of all 496 species ever recorded in Finland, it contains all breeding species, non-breeding migrants and most common vagrants, making it unlikely that a citizen records a species not included in the classification model.”

Lines 622-623 and 627-628

I would remove the phrase ‘conditional on ...’. Isn't this condition already implied in the formula - when m_j equals 1 and not 0?

We agree that it is implied by the formula, but we have left the ‘conditional on ...’ to further clarify this in the text.

Chapters on prior models

Where can I find the R-scripts used for generating the prior models? Am I correct to assume that the coefficients of the priors are fixed parameters in the DT?

We have added the R-scripts for generating the prior models in the Zenodo repository. The reviewer is correct that the hyperparameters (coefficients of the prior) are fixed in the DT.

Chapters on posterior models

I am not sure why the sequence of the chapters is inconsistent with the chapters on the priors. I must admit that I find this section hard to read and therefore difficult to review. Being able to see the code may help in my understanding but also in replicability of the study.

We have now reorganized the prior model descriptions to follow the same ordering as the descriptions of the posteriors, since the latter are naturally ordered according to the order of

model component fitting. We have also rewritten these descriptions to be more intuitive and accessible, and added the code in the Zenodo repository.

Computational implementation chapter

I find this chapter very useful. Thank you for this!

We thank you for the positive feedback.

Reviewer #2 (Remarks to the Author):

Review for NATECOLEVOL-25072566-T:

“A Digital Twin for Real-Time Biodiversity Forecasting with Citizen Science”

Reviewer Background: PhD-Level applied statistician with experience in statistical learning and causal machine learning methods for population monitoring data. Active areas of research include statistical learning models for animal movement and epidemiological applications for wildlife populations, causal machine learning methodology for trend estimation from structured and opportunistic survey data.

Overall Comments:

The authors present a new data collection app (referred to as the MK app) and methodology (referred to as Digital Twinning [DT]) to predict and forecast species distribution.

Overall, I have several issues with this work that I feel need to be addressed. Terminology is poorly defined and utilized throughout the manuscript. I am not convinced that a general audience at Nature Ecology and Evolution could understand the arguments made and presentation of methodology based on the current presentation. I believe this stems from the author's not situating this work with the extensive literature on data integration and species distribution modeling and forecasting. I did not find the presentation of the methodology accommodating of a general quantitative audience that would want to understand modeling decisions made. These issues I have explained in greater detail in the following sections.

We thank you for your critical and constructive comments, which were highly valuable and resulted in major rewriting of many parts of the manuscript. We address them in the following sections, where the Reviewer has explained each of these comments in greater detail.

Pertaining to the results and discussion, one major concern I have is the lack of commentary on statistical or model-based inference from their model. Environmental variables were used to help estimate species distributions, and the relationships between current environmental conditions and predictions of future species distributions are important. The authors did not provide enough information about the model to help me understand if there is potential for inference about relationships between environmental conditions and future states of a population. As an example, if forecasted distributions decrease in size or detection probability year-after-year can this model help explain potential drivers of this change? If this inference is not easily attainable, it would be a bit of a disappointment and worthy of a disclaimer. I don't

think this would be a deal breaker for using methodology such as this, but it would more accurately avoid portraying this work as a panacea.

This is an important point, and we thank you for pointing out the lack of clarity. We have now clarified that environmental variables were used to estimate species distributions in the prior model, but the updating from the prior model to the posterior model uses only the MK app data and spatial smoothing:

”The updating of the spatial distribution component is conducted directly at the level of the model predictions through spatial smoothing, not at the level of prior model parameters that map for instance the environmental affinities of the species.”

Furthermore, as we now point out in the Discussion, while the modelling approach does not directly infer potential drivers of changes in species distributions:

“While the DT developed here is targeted at quantifying changes in species distributions rather than directly inferring potential drivers of such changes or suggesting management or policy actions, it forms the basis for taking informed steps into these directions.”

"Predictive performance is a reliable way to assess models such as this, but I think the authors should make more effort to explain the qualitative challenges of using information from this methodology for policy development. I think the relationship between model interpretability and potential to inform policy is frequently overlooked.

We agree and have clarified that the DT does not suggest management or policy actions. In theory, the improved SDMs after DT could be used for identifying habitat management sites for threatened species, such as in forestry, agriculture or wetland management purposes. However, such kind of policy tools (or strong will towards that) are not yet in place in Finland. We hope that the nature restoration law may put some pressure on this in the future. Unfortunately, we were not able to elaborate on these aspects in the manuscript, as the reviewers suggested so many valuable additions that we faced challenges in staying within the journal's length limits.

Further, I was a bit disappointed to find misleading comparison of the MK app to the eBird project, which is arguably one of the largest and most successful intercontinental citizen science projects in human history in several aspects. This is both from a data perspective (i.e., data volume, data quality, quality control, quantification of user skills through checklist calibration indices) and methodology for citizen science data (i.e., accounting for confounding and changes in the observation process, rigorous statistical simulations, external validation with other structured survey data, etc.). There are some statements made in the discussion that I personally felt were incorrect or had overstatements of the MK app relative to the eBird project. It would be impossible for the MK app to accomplish what eBird has become over the past two decades in a matter of 2-3 years. My advice to the authors would be to limit mentions of eBird or to better familiarize themselves with what has been accomplished with the eBird project in order to more appropriately contextualize the new work with the MK app. The MK app seems like an exciting national-level citizen science project, which I would be excited to see grow further.

We sincerely did not intend to present any misleading comparisons or to downplay the success of the eBird project, which we fully agree has been one of the largest and most successful intercontinental citizen science initiatives. Our intention was solely to contextualize the new work with the MK app by explaining how it differs from eBird. To achieve a well-balanced discussion, we have now carefully revised the section comparing our approach to eBird :

“Citizen science can provide massive amounts of biodiversity data. For example, the platforms eBird³⁹, iNaturalist⁴² and Pl@ntNet⁴³ have recruited some 1.1M, 8.9M and 8.2M users respectively. These massive citizen science data have not only provided an invaluable resource for biodiversity research but also inspired a large body of statistical methods development needed to account for data quality issues such as sampling biases and detection errors²³. For example, although eBird’s data collection procedures involve systematic quality control and quantification of user skills, using these data for prediction and inference requires statistical approaches that carefully account for confounding factors and changes in the observation process. The best practice recommendations for using eBird data involve choices related to filtering the data for complete checklists, performing spatial subsampling, and using filters for observation effort⁴⁴. The predictions based on eBird data that we utilized in our comparison are not updated automatically in real-time, but periodically by Cornell Lab of Ornithology data scientists, who provide Status and Trends products based on data accumulated over several years.”

Furthermore, following a suggestion by Reviewer 1, we now provide a comparison to eBird-based predictions. We describe the results of this comparison in the main text as follows:

“We further compared the DT predictions to those based on the eBird³⁹ global citizen science project. We extracted for each survey week species occurrence probabilities from eBird Status and Trends Weekly Abundance Maps released in summer 2025, representing data that have accumulated until 2023⁴⁰. For those 53 species for which eBird-based predictions were available, mean AUC was 0.62 for eBird-based predictions and 0.67 for DT predictions.”

Summary/Abstract:

2nd Sentence: Are the issues related to data quality or variance in data quality? As a statistician who develops machine learning models for eBird project data and other more structured surveys (e.g., North American Breeding Bird survey, Saltmarsh Habitat and Avian Research Program, etc.), I am suspicious that the issues is likely variance in data quality. Within the eBird project, there is an exceptional amount of high quality data, but there is also a lot of lower quality data mixed in. I cannot speak for this citizen science platform, but generally for eBird, variance in data quality is the main issue that we seek to address in any modeling problem.

Thanks to the extensive comments by the Reviewer, we have now better identified the novelty of our work. This resulted in major rewrite of the first half of the abstract, which now reads as follows:

“Citizen science provides massive amounts of biodiversity data, yet its full potential remains untapped due to two challenges: how to meaningfully involve also less skilled citizens, and how to accelerate the process from data collection to research and monitoring outputs. We show how even those citizens who cannot identify birds themselves can substantially contribute to

real-time predictions on bird distributions. This is achieved through a Digital Twin (DT) that combines smartphone-based citizen science with long-term knowledge in a continuously updating model. The app submits raw audio to backend that classifies birds with machine learning, reducing variation in data quality and enabling validation and reclassification by continuously improving classifiers.”

Remaining sentences: I am not 100% sure who the audience is based on this summary in the final sentences. Digital Twinning is a term that I suspect is more familiar to engineers and computer scientists. Coming into reading this paper, I have no idea what this term meant, I have read the definition in the introduction several times and to be honest I didn't understand it until I read the whole paper closely multiple times. I think what makes this confusing is that the fields of conservation ecology, wildlife management, and environmental statistics use the term “data integration” for statistical and/or machine learning modeling uses various sources of data to improve inference. See the following references for some examples:

Zipkin, E. F., Zylstra, E. R., Wright, A. D., Saunders, S. P., Finley, A. O., Dietze, M. C., ... & Tingley, M. W. (2021). Addressing data integration challenges to link ecological processes across scales. *Frontiers in Ecology and the Environment*, 19(1), 30-38.

Zipkin, E. F., Inouye, B. D., & Beissinger, S. R. (2019). Innovations in data integration for modeling populations. *Ecology*, 100(6), 1-3.

I have more comments on this in the introduction.

We now cite these two papers in the Introduction. We also address this topic in the later comments where the reviewer develops the discussion on terminology (especially digital twinning versus data integration) further.

Main:

-In the second paragraph, it may not hurt to have this citation? I'll leave the authors to decide if citing this manuscript is needed here. A large aspect of the potential of citizen science data research in the U.S. is rooted in semi-structured protocols and the documentation of various aspects of the observation process associated with each observation.

Steve Kelling, Alison Johnston, Aletta Bonn, Daniel Fink, Viviana Ruiz-Gutierrez, Rick Bonney, Miguel Fernandez, Wesley M Hochachka, Romain Julliard, Roland Kraemer, Robert Guralnick, Using Semistructured Surveys to Improve Citizen Science Data for Monitoring Biodiversity, *BioScience*, Volume 69, Issue 3, March 2019, Pages 170-179, <https://doi.org/10.1093/biosci/biz010>

We thank you for the suggestion and have now added this citation to the paper and extended the sentence as follows:

“As it is difficult to reliably account for the variability in citizens in their skills of identifying species, as well as to quantify the spatiotemporal sampling effort, it remains hard to disentangle biological signals from these observation biases, especially if sampling effort is not carefully documented^{25,26}.”

-“Digital Twinning (DT), i.e., the pairing between real-world objects and their digital representations, has been much used in technology to speed up development cycles and to make them more cost-efficient²⁶. While originally developed to simulate engineered systems, there is widespread interest in applying DT to other fields, including biodiversity research. By combining data, models and domain knowledge in a close to real-time alignment with the real world, DT holds considerable potential in ecological forecasting and in enabling rapid environmental decision-making^{27,28}....”

With no offense intended, these 3 sentences mean virtually nothing to me. As a reader, I don't know anything about digital twinning, I don't understand how the term “real world objects” connects to citizen science and I don't know what a “digital representation” of a real world object is. My impression reading this is that domain-experts on this topic carelessly dumped these technical terms into an introduction for an ecology readership. After reading the whole manuscript carefully multiple times and rereading these sentences, I am pretty sure I know what the authors mean here, but requiring this level of effort is not considerate of readership. For a Nature Ecology and Evolution manuscript, I suspect the editorial board will appreciate the author's more carefully considering the audience, the language readers are familiar with, and how the work is closely related data integration work of others. How does digital twinning differ from data integration? Are they the same topic with different names??

The concept of digital twinning is indeed quite new in ecology, and one aim of our manuscript is to introduce this concept to the audience. We now explain in several parts of the introduction how digital twinning in ecology differs from data integration (or any type of modelling):

“Digital Twinning (DT) refers to the concept of creating a digital counterpart of a real-world system. In ecology, DT could mean building a dynamically updated digital model of a species' distribution or an ecosystem's state, based on continuously incoming observational data. While originally developed in engineering to simulate and optimize physical systems²⁷, DT is gaining interest in biodiversity research, where it can help integrate data, models, and expert knowledge in near real-time²⁸⁻³⁰. This approach holds promise for improving ecological forecasting and supporting timely environmental decision-making³¹.”

“We build on recent approaches in data integration^{32,33} and integrated species distribution modelling³⁴ to combine the continuous flow of new citizen science data with previous long-term data on bird spatial distributions, timing of migration, and patterns of singing activity.”

“A core feature that distinguishes a DT from merely data integration, is that the DT goes further by maintaining a dynamically updated model that mirrors the real-world system as it evolves over time, here distributions, migrations and singing activity of birds, as well as citizens recording them. By relying solely on machine learning based bird classifications rather than citizen-based classifications, we remove an important part of observer heterogeneity and increase inclusivity by enabling ordinary citizens without bird identification skills to take part in data collection by making bird recordings.”

A more elaborated discussion on this topic, in the ecological context, is given in the two references that we cited in the earlier version of the manuscript (de Koning, K. et al. 2023. Digital twins: dynamic model-data fusion for ecology. *Trends Ecol Evol* 38, 916–926; Lecarpentier, D. et al. 2024. Developing prototype Digital Twins for biodiversity conservation and management: achievements, challenges and perspectives. *Res Ideas Outcomes* 10) as well as in a newer references that we have now added (Khan et al. 2025. TwinEco: A unified framework for dynamic data-driven digital twins in ecology. *Ecological Informatics* 91, 103407). We now further cite Buerk et al. 2024 which demonstrates how digital twins can be used in a policy setting.

Now to the next sentences:

-“... However, the development of digital twins for biodiversity remains a complex and emerging research frontier, hindered by the complexity of natural ecosystems, the need to combine heterogeneous data sources and the technical challenges associated with generating and processing real-time biodiversity data streams.”

Combining heterogeneous data sources would be considered data integration to almost any researcher in the Ecological Society of American (ESA) or the British Ecological Society (BES). If you choose to use a different term for the exact same concept, you risk confusing readership. If you choose to use a different term for a slightly different concept, the differences need to be clearly explained somewhere.

We agree that combining heterogeneous data sources is data integration, and we have now clarified this in the manuscript. However, as we explain above, data integration is only part of a digital twin.

Also, what is a “data stream?” Do all ecologists know what a data stream is? What is the difference between a data stream and a data collected from a designed study, opportunistic data from citizen science projects, or data from audio recording units? I know what a data stream is because I have collaborated with computer scientists and data scientists, but I am not sure other readers will immediately know what you are talking about here.

We have found the term data stream to be much used in the recent literature and consider it appropriate here. In our view, the term data stream emphasizes the continuous nature of data collection (and processing).

-“This paper aims to demonstrate the applicability of DT approaches in biodiversity research for achieving accurate, real-time predictions of species distributions.”

Integrated species distribution modeling literature seems to be disregarded in this paper:

Ahmad Suhaimi, S. S., Blair, G. S., & Jarvis, S. G. (2021). Integrated species distribution models: A comparison of approaches under different data quality scenarios. *Diversity and Distributions*, 27(6), 1066-1075.

Koshkina, V., Wang, Y., Gordon, A., Dorazio, R. M., White, M., & Stone, L. (2017). Integrated species distribution models: combining presence-background data and site-occupancy data

with imperfect detection. *Methods in Ecology and Evolution*, 8(4), 420-430.

Dovers, E., Popovic, G. C., & Warton, D. I. (2024). A fast method for fitting integrated species distribution models. *Methods in Ecology and Evolution*, 15(1), 191-203.

Forester, B. R., DeChaine, E. G., & Bunn, A. G. (2013). Integrating ensemble species distribution modelling and statistical phylogeography to inform projections of climate change impacts on species distributions. *Diversity and Distributions*, 19(12), 1480-1495.

Buisson, L., Thuiller, W., Casajus, N., Lek, S., & Grenouillet, G. (2010). Uncertainty in ensemble forecasting of species distribution. *Global Change Biology*, 16(4), 1145-1157

Lawler, J. J., Wiersma, Y. F., & Huettmann, F. (2010). Using species distribution models for conservation planning and ecological forecasting. In *Predictive species and habitat modeling in landscape ecology: Concepts and applications* (pp. 271-290). New York, NY: Springer New York.

This leads to a major comment. The authors seem to have disregarded literature in ecology on this subject and have made no clear effort in the introduction to connect/situate their work relative to existing and ongoing research. I am an advocate for bringing new methods and ideas to the table in an interdisciplinary setting. However, this paper so far does a disservice to the reader by not helping contextualize the proposed methodology more broadly.

We thank you for the references provided, of which we now cite Suhaimi et al. (2021). As noted above, we have added a substantial amount of text to clarify the relationship of our work and integrated species distribution modelling.

-“This paper aims to demonstrate the applicability of DT approaches in biodiversity research for achieving accurate, real-time predictions of species distributions. We illustrate this through a case study in audio-based bird monitoring, showcasing how reliable real-time biodiversity predictions can be achieved through a DT approach that combines the strengths of citizen science, machine learning, and high-performance computation. Our approach features a continuous model updating process, ensuring that predictions remain responsive to real-time changes in bird activity and environmental conditions. The technological innovations developed in this study not only reduce the time required to generate accurate biodiversity information for policy and management, but also increase inclusivity by broadening the stakeholder community and the roles of the stakeholders. This approach empowers and engages citizens to provide pivotal contributions to both scientific research and environmental monitoring.”

I have mixed feelings about this paragraph. It has a lot of good information that pulls me in as a reader. However, I am left more confused by this paragraph than before I started. The underlined section is where the confusion begins for me. In this paragraph, there is a first mention of “a case study in audio-based bird monitoring. Audio-based bird monitoring is often NOT citizen science data. Is the citizen science data different than the audio data? Are the same thing? If there are two sources of citizen science and/or non-citizen science then what are they? Are we then integrating audio data with another data source? HELP! I am lost.

Even if the authors clarify later on, this lack of clarity is not appropriate for a Nature Ecology and Evolution manuscript. Any and all data sources that will be integrated within the DT framework should be clearly listed here to help the reader understand what data sources the authors use to estimate and forecast species distributions.

As clarified in the revised text, our specific case study involves audio-based monitoring carried out by citizen scientists using the MK app (which is an automated bird sound identification app), so the citizen science data are the same thing as the audio data. We integrate this new citizen science data with other data sources which relate to long-term bird monitoring. Those data sources are described in the section that explains the prior model. The MK app data (=the citizen science audio monitoring data) are then used to update this prior model to a posterior model. These comments prompted the major rewrite of the first part of the abstract, where we now better explain the novelty of our work and the approach taken:

“Citizen science provides massive amounts of biodiversity data, yet its full potential remains untapped due to two challenges: how to meaningfully involve also less skilled citizens, and how to accelerate the process from data collection to research and monitoring outputs. We show how even those citizens who cannot identify birds themselves can substantially contribute to real-time predictions on bird distributions. This is achieved through a Digital Twin (DT) that combines smartphone-based citizen science with long-term knowledge in a continuously updating model. The app submits raw audio to backend that classifies birds with machine learning, reducing variation in data quality and enabling validation and reclassification by continuously improving classifiers.”

In addition to explaining all data sources in the text, we have summarized them in the revised version of Figure 2:

A new tool for digital citizen science:
 -finetuned should be written as fine-tuned

Thank you – revised as suggested.

–“First, to mitigate the differences in species identification skills among citizens, all classifications are performed by a machine learning model, with users given the option to confirm or reject the classification.”

To be honest, I find this measure of observer skill to be unsettling. With any species identification application, the quality of the recording, the species family, and environmental conditions can have substantial consequences on classification performance. Observer skill should be assessed relative to other observers. Also what does “reject the classification” have to do with observer skill? Did they reject because they didn’t see the species (and they want to see it to add it to a “life list”)? Did they reject because they know the application is not classifying the species correctly? Did they reject because they did not have the ability in that one instance to follow-up with a confirmed visual of the individual? Did they reject because another individual in their party used playback calls, which were subsequently picked up on a audio recording. I would hope that other reviewers have a second opinion on this topic, but I do not think basing observer skills off a sound ID app is appropriate. There are other groups working on metrics for observer skills. An example is the checklist calibration index, developed by a group of researchers at Cornell Lab of Ornithology, which measures the relative skill of an observer skill at identifying/finding x number of species for t minutes of search effort and d meters traveled. Without more detail about how observer skill was measured under this approach, I cannot recommend this work for publication. How observer skill is measured is

arguably one of the most important aspects of any citizen science research, and the researchers have made a controversial choice (which is not inherently bad) and then avoided justifying or explaining how this choice is ok.

As the reviewer rightfully points out, variation among users in their skills and behavior has many dimensions. In the revision, we have amended the text in many places to better discuss dimensions that remained unclear in the original version.

In the MK app, all detections are based on the user recording audio (with three different methods, as detailed below), and then submitting the audio to our backend from where a machine learning model returns classifications to the user's phone. We clarify this in the revision as follows:

“First, to mitigate the differences in species identification skills among citizens, all classifications are performed by a machine learning model and thus bird identification by citizens is not required.”

Each classification is associated with a probability score (see below about how the scores were calibrated to probability scale), which is shown to the user as such, and furthermore color-coded as green ($\geq 90\%$) or orange ($\geq 30\%$ and $< 90\%$). The user has the option to confirm or reject each such classification, which we encourage the user to do especially if they think the classification was incorrect. This may be the case if, for example, a user who knows birds well has seen the bird or listened to its vocalization. However, our earlier work (Nokelainen et al. 2024, cited in the manuscript) has shown that while some users provide accurate corrections, these corrections vary a lot among users. For this reason, we have not accounted for the user corrections in the modelling conducted in this paper but rely only on the machine learning based classifications. We mentioned the possibility of the user rejecting or accepting a classification as one of the features of the MK app, but thanks to the reviewer comment we realized that this could result in confusion. Thus, we have removed any mention of the user corrections, making it clear that we only rely on machine learning based classifications. This is our main argument of how we mitigate the differences in species identification skills among citizens: their bird identification skills do not influence in any way the process of classifying species from the audio. What variation among users however influences is when and where they record birds. We now discuss this explicitly:

“Despite the above-mentioned features, the MK app data has some of the biases that are characteristic to citizen science datasets. Most importantly, the direct recordings are triggered by bird vocalizations that are of interest to the users. As shown in our previous analysis, some users target only new species that they have not recorded before, whereas other users provide data that are comparable to passive audio monitoring⁴⁵. Accounting for such variation in user profiles provides an important challenge for future work. Another limitation is that the MK app is based on audio only, omitting visual observations of birds.”

Another aspect that mitigates heterogeneity and biases related to species classification is that we have calibrated the classification models specifically to the MK app data, as was explained by the methods. This calibration is made possible by the fact that we store all raw audio at the backend (not just the classification results). This allows us to manually evaluate the quality of the machine learning based classifications.

-“Second, to mitigate spatial observation bias and preferential sampling, the MK app enables not only direct recordings, but also interval recordings and systematic point counts.”

In my opinion, this is extremely poor writing and completely glosses over an introduction of a second data source! I think this section needs to be rewritten. If there are three data sources, the three data sources should leap off the page and scream at the reader. A numbered list of the 3 data sources would be ideal. Also, after reading this paragraph, I don't understand how interval recording mitigates preference sampling or observation bias? Couldn't observation bias and preferential sampling be made even worse? I don't think anyone is leaving a phone recording in a park or random location... I would think that almost every sensible adult would participate in interval recordings near home or some secure location. I am suspicious that the “mitigation of bias and preferential sampling” refers to the systematic point counts.

We thank you for pointing out that our writing was not as clear here as it could have been. We have followed your advice and first listed the three data sources and then clarified their pros and cons. We have added the following sentences to the text:

“The MK app includes three recording types: (1) direct recordings, (2) interval recordings, and (3) point count recordings.”

“While the interval recordings do not remove the spatial bias of where the recordings are conducted, they largely remove the temporal preferential bias of when they are conducted. Even if the initiation of an interval recording would be triggered by bird vocalization activity, after the first 9-minute break, the recorded minutes represent bird vocalization activity in much less biased way than direct recordings”

“The permanent point count locations mitigate spatial observation bias, as the citizens make recordings at pre-selected locations. They also partially mitigate the temporal bias, because the recording interval is five minutes long, and thus especially its latter part is less dependent on whether bird vocalization activity triggered the initiation of the recording. We have furthermore encouraged users to initiate point count recordings whenever they walk through the route, disregarding whether birds are vocalizing or not.”

With regards to mitigating preferential sampling and observation bias, I cannot say I am convinced. I would strongly suggest fitting distribution models with each source independently before any type of “twinning” or integration”... (still not 100% what the authors are doing exactly). From what I have read so far, I'd be much more conservative with regards to any claims of mitigating bias and issues with preferential sampling unless they can show that the integrated/twinned estimates are improved over disjoint estimation of species distributions. This could be done through simulation but it could also be done through external validation with historical estimates of abundance.

While we still consider that the reliance on machine learning based classifications in the MK app helps to mitigate variation in user's bird identification skills, we agree that preferential sampling is an important issue. Thus, we have added more thorough discussion of this aspect (as described above). The reason why we have not fitted the model separately to each data type is that the amount (and especially spatial representativeness) of the interval and point count recordings is much smaller than of the direct recordings. Thus, using them alone would not be sufficient to finetune the predicted distributions with high spatial resolution. Instead, we

account for the difference among the three data types in the detection model, which includes data type as one of the covariates. We note that the validation against independent data shows that our approach results in better predictions than predictions generated by the prior model. To what extent this is achieved through each of the three data types remains to be seen in the future, once we have generated sufficient data of each type.

A real-time biodiversity Digital Twin:

-“We developed a Digital Twin (DT) that predicts spatiotemporal distributions of bird occurrences and their vocal activity across Finland, with a spatial resolution of one-hectare and a temporal resolution of one hour.” I am so sorry. I still don’t know what a digital twin is and how you develop one. (Maybe I’m incompetent and if so I apologize for being assigned as your reviewer.)... After multiple hours, here is my guess: Is DT literally a predictive machine learning model that is fit to multiple data sources to estimate species distributions as a probability of occurrence? If this is true, why not say this? Ecologist should understand this language as it is in-line with ecological statistics literature and standard statistical learning literature.

As noted above, we have now added a definition of a digital twin to the introduction. We have also explained how it differs from a model, which it incorporates as one component. We acknowledge the challenge of achieving a terminology convenient for all readers despite their different backgrounds. From this perspective, we were pleased to observe that Reviewer #1 (with expertise in “digital twins, Bayesian statistics, modelling”), and Reviewer #3 (with expertise in “bird population monitoring”) appeared to be comfortable with the terminology used in the original version. We hope and trust that our renewed attempt at further integrating our work with the terminology of data integration and integrated species distribution modelling will help other readers to relate our work to their knowledge base.

-“We quantified prior knowledge on bird species’ spatial distributions by fitting the joint species distribution model HMSC31 to long-term data on transect-line surveys, using as predictors one-hectare resolution raster maps of land-use variables, forest structure variables, and climatic variables (Fig. 2B).”

HMSC??? Is that an acronym for some sort of ‘Hidden Markov State-Space Model’? Perhaps I have missed acronym from earlier? I am still very confused about the Digital Twin. Is the twin the joint species distribution model? Is it something else?

Thank you for pointing out that we had failed to spell out in the main text that HMSC stands for “Hierarchical Modelling of Species Communities”, which is one specific joint species distribution model. We now include this information in the main text. As mentioned, we now explain further what the twin is, while stressing that it is not identical to the joint species distribution model (which is merely one part of the prior model).

Beyond this, perhaps the authors were discouraged from including any equations in this work, but I am at a loss for how to evaluate this model. I keep trying to sort through various what variables are being fit to what portions of the model and what model fitting challenges may arise. It may all be valid, but I don’t have the time to figure it out as a reviewer, which I apologize for. I can’t say that Figure 2 helps me as much as I had hoped when I saw the reference.

We are a bit unsure of what the reviewer means by “...from including any equations in this work...”, as in the Methods section we explain the modelling approach in great detail in terms of equations. This is done in particular in the sections “Overview of the Digital Twinning approach”, “Prior model for migration”, “Posterior model for detection”, “Posterior model for migration”, “Posterior model for spatial distribution” and “Evaluation of predictive performance using expert point count data”. We have attempted to keep the main text as readable as possible for the general audience and thus avoided equations there. We also note that explaining our approach satisfactorily in terms of equations requires substantial space. This is evidenced by the fact that the Methods sections referred to above spans over more than 13 manuscript pages. Thus, excluding equations from the main text is not only a question of style, but also a question of length restrictions.

Generally speaking, I have some concerns about estimation of species distributions during any migratory period. It makes me extremely uncomfortable. Some species are vocal during migration and some are not... I am not an ecologist so I don't fully understand the utility of real-time forecasting. Also, when I start to think of species that are detected visually because they are primarily non-vocal (which there are many), I also start to feel less comfortable with what is being presented here.

As explained both in the main text and the Methods, our prior and posterior models are products of three components, which model (1) whether the species is present at the latitude of observation from the migratory point of view, (2) whether (conditional on 1) it is present in the habitat where the recording is conducted, and (3) whether (conditional on 1 and 2) it vocalizes at the time of the recording in a manner that leads to a detection. Thus, migration period bird vocal activity (or the lack of it) are controlled in our detection model. Since citizens tend to be particularly interested in bird migration, especially in the arrival of birds in the spring, we are strongly motivating to keep this component as part of the twin. However, we fully agree that the fact that the MK app ignores visual detections is an important limitation, which we now address in the discussion: “Another limitation is that the MK app is based on audio only, omitting visual observations of birds.”

I am happy to give another go at understanding what the authors are presenting here, but I do not have time to sort through this in it's current state, especially with no frame of reference regarding the similarity or lack thereof between DT and data integration. Also, I keep getting “MK app data” mixed up with “MK app recording.” Are the transect-line surveys recorded within the MK app data? Or are they something different. I am starting to wonder if it would be good to have a glossary table of definitions and terms.

As mentioned above, we have now included in the Introduction a section relating digital twinning to data integration. We have clarified that the data used in the prior model (such as transect line surveys) are independent of the MK app data. We refer by “MK app recording” to an individual recording made by a user, and by “MK app data” to the collection of all data produced by the app (recordings classified by the AI model). These are indeed closely related as all data from the MK app are based on the recordings.

Example predictions:

No comments for now.

Evaluation of predictive capacity:

-Why is next day performance the metric for assessing predictive capacity? What about next week performance? Next month or next year performance? Is it because the species are not in the MK study area 1 week to 1 month out? My opinion on this should not be the authority, but I was hoping to see longer term forecasting performance assessments.

We acknowledge that selection of next day performance as the metric is somewhat arbitrary. We could equally well have selected next week or next month performance. Given that the app has been operational only for a few years, evaluating next year performance is not yet feasible. The specific selection of next day performance was motivated by what we considered the most relevant question from the user's point of view: how well the most updated predictions inform us about where the birds are right now.

-“While the above-described evaluation of predictive performance was based on splitting the data temporally into training (MK data until present day) and testing (MK data for the next day), the test data were not fully independent from the training data for two reasons. First, the same citizens may have recorded the same bird individuals on consecutive days. Second, the machine learning based classifications may contain consistent mistakes both in the training and in the test data, potentially inflating the apparent predictive performance.” I do not understand why counting the same bird on consecutive days matters for performance assessment. For resident birds, this is going to happen and it is a critical characteristic of species distributions. If this is really important, I think the authors should clearly explain why this is an issue.

We agree that this is a critical characteristic of species distributions. However, it also makes the test data not fully independent of the training data. We have now included this reasoning in the Methods:

“While the above-described evaluation of predictive performance was based on splitting the data temporally into training partition (MK data until present day) and unseen testing partition (MK data for the next day), the test data were not fully independent from the training data for two reasons. First, the same citizens may have recorded the same bird individuals on consecutive days. In such a case, the updated model would be likely to predict that the individual will be there on the following day, too. Through spatial smoothing, the updating will influence predictions also on whether the species is present somewhere nearby. To test whether this does not only influence but also improve predictions in those nearby locations, fully independent test data are needed.”

You have lost me again... What do you mean by “the machine learning based classification may contain mistakes.” This sentence is utterly useless because all machine learning models contain classification mistakes. If they don't then they are almost certainly overfitting to the data. Are these misclassifications of species based on audio recordings? Or are you referring to the machine learning model that you are using to fit a model to the trained data? I suspect that this second sentence is going to confuse people more than it will help.

The reviewer is correct that we specifically refer to misclassifications of species based on audio recordings. We have added the following clarification:

“Second, the machine learning based classifications may contain consistent mistakes both in the training and in the test data, potentially inflating the apparent predictive performance. To see why this could be the case, assume that MK app detections of species A would be based on misclassification, as in reality originating from species B. Validation against next day MK app data could yield optimistic results, because both the predictions and the next day data might agree that A is present, even if this was not the case. Instead, manual point counts by bird experts would correctly suggest that species B is present instead of species A and thus avoid circularity in validation.”

We note that while this consideration motivated the independent validation experiment with manual point counts, the results showed that validation results were very similar to the MK app data, suggesting that such potential biases were not common in our data.

-“To evaluate the difference between the posterior and prior predictions against fully independent data, we further performed manual point counts by bird experts in pre-selected locations from May 7st to June 7th, 2025.” What is a “bird expert?” Is this a ornithologist? An seasoned volunteer bird-watcher?

We have added this clarification: “The bird experts were seasoned volunteer birdwatchers, whose capacity to identify birds from their vocalization has been demonstrated, e.g., by providing high quality survey data to the national line transect or point counting schemes.”

-“The manual point count locations were selected algorithmically to include sites where the prior and posterior predictions were as contradictory as possible.” If this is a form of external validation (using completely independent data), then why does this choice matter? Doing external validation with strong algorithmically-assigned preferential sampling sounds odd to me. This should be justified.

To justify, we have added the following section to Methods:

“This algorithmic approach was used to increase the information content of the data. For example, consider a region from where there are no (or only few) MK app recordings. The lack of local data would make the prior and posterior predictions identical, and hence their comparison through validation not meaningful. As the bird experts conducting manual point counts were unaware of the prior and posterior predictions, the algorithmic selection of the validation sites does not bring bias to the results.”

Discussion:

2nd paragraph- I am not a fan of this paragraph, particularly the manner in which eBird is juxtaposed to the MK app.

-“However, the use of these massive citizen science data resources for scientific inference has remained challenging, in particular due to data quality issues such as sampling biases and detection errors²³. For example, the best practice recommendations for using eBird data involve choices related to filtering the data for complete checklists, performing spatial subsampling, and using filters for observation effort³⁶.” Even structure surveys such as the

North American Breeding Bird Surveys has sampling biases (e.g., roadside surveys have road side survey bias) and detection errors. Human participants or remote detection equipment will inherently fail to detect some species. Complete checklists in eBird have all information regarding location, time, day or year, effort distance, effort hours, number of observers, stationary vs. traveling, and although it is not currently used in most general applications, the eBird project contain GPS tracks for a large majority of checklists which provides important information about how far search effort disperses from the recorded location of the checklist. The eBird project data is completely independent of the Merlin Sound ID app (until recently). The eBird project also contains data from more structured survey efforts such as breeding bird atlas projects. With eBird data, observer effort can be quantified through a “checklist calibration index.” There are several papers regarding method development for eBird data that utilize this index.

We thank the reviewer for pointing us to papers regarding method development for eBird data that utilize the checklist calibration index. Indeed, what we attempted to say in this section is that almost any citizen science data (such as North American Breeding Bird Survey or eBird) have sampling biases, which need to be accounted for in the analyses, for which sophisticated methods have been developed. We also stressed that while the MK app data were specifically developed to overcome some of the sampling biases (such as variation in observer skills in bird identification), they also suffers from preferential sampling and some other biases. We have now attempted to make this discussion more balanced; in particular, we have nuanced the manner in which the MK app is compared to eBird:

“Citizen science can provide massive amounts of biodiversity data. For example, the platforms eBird³⁹, iNaturalist⁴² and Pl@ntNet⁴³ have recruited some 1.1M, 8.9M and 8.2M users respectively. These massive citizen science data have not only provided an invaluable resource for biodiversity research but also inspired a large body of statistical methods development needed to account for data quality issues such as sampling biases and detection errors²³. For example, although eBird’s data collection procedures involve systematic quality control and quantification of user skills, using these data for prediction and inference requires statistical approaches that carefully account for confounding factors and changes in the observation process. The best practice recommendations for using eBird data involve choices related to filtering the data for complete checklists, performing spatial subsampling, and using filters for observation effort⁴⁴. The predictions based on eBird data that we utilized in our comparison are not updated automatically in real-time, but periodically by Cornell Lab of Ornithology data scientists, who provide Status and Trends products based on data accumulated over several years.”

-“A core feature of the MK app is that it was directly developed to overcome the outstanding challenges of citizen science²³. First, to tackle the issue of variable and often unknown sampling effort, the MK app quantifies the location, time, type, and duration of each recording, and implements standardized interval recordings and permanent point count routes. Second, to remove observer heterogeneity in species identification, the MK app uses machine learning-based classifications with well-calibrated estimates of uncertainty. These characteristics of the MK app data facilitated their straightforward integration into a predictive DT approach without the need for statistically controlling for sampling artefacts.”

-artefacts ... should be artifacts?

Thank you for noting that our language was not fully consistent in terms of whether we used British or American English. We have now ensured that we consistently use British English and have thus kept “artefacts”.

-The first and second sentence of this quote makes me wonder if the authors have read the Kelling et al. 2019 paper mentioned previously? The authors are selling the MK app as if it has accomplished something (regarding overcoming the challenges of citizen science) that eBird hasn't. I don't think it is true. Reading this work does not convince me that this statement is appropriate as well. The third sentence is even more concerning. The authors have NOT clearly explained how the MK app uses machine learning based classifications to remove observer heterogeneity in species identification. This is a big and controversial idea (especially if it relates to using audio recordings to judge the skill level based on what users end up reporting for a list of species during a given search effort). The authors provide no justification or evidence that this is appropriate and I could not find any mention of appendices where this choice was rigorously evaluated.

As discussed above, we have now clarified that we only rely on machine learning based classifications. This removes observer heterogeneity in species identification, because the identification is not based on observer skills but on an machine learning based bird classifier, as described now in the abstract, main text and methods. Importantly, this approach extends the user base, allowing anyone to become a citizen scientist. This is an important step forward, as a large proportion of our users do not have any skills in bird classification, at least when they started using the app (as evidenced by the fact that 5% of the Finnish population uses the app). We cite the Kelling et al. 2019 paper in the revision. We do agree that eBird provides an excellent platform, an outstanding tool and probably the most widespread citizen science initiative on birds. However, its key difference to our initiative is that while data integration focuses on combining datasets from different sources (such as in eBird), Digital Twinning goes further by maintaining a dynamically updated model that mirrors the real-world system as it evolves over time.

-I can understand why the comparison to eBird could be discussed in this paper, but this comparison really falls short. I think part of it is caused by the fact that the MK app is newer and records observations directly through audio recording and through other forms. The eBird app does not directly allow for audio files to be converted into checklists. There is a newer way to do that from the Merlin Sound ID application but this does not represent a large portion of the data collected over the past +20 years. See also my major comment in the Overall Comments section about interpretability and inference.

As discussed above, we have made the comparison to eBird more balanced. As a response to Reviewer #1, we have also included an explicit comparison to eBird-based predictions.

Reviewer #2 (Remarks on code availability):

The paper needs to be more clearly written and contextualized with existing literature before any

further work by a reviewer.

We hope and trust that the revised version is clearer in its writing and contextualization, and thank the reviewer for pointing out several aspects that remained unclear for them in the original version.

Reviewer #3 (Remarks to the Author):

This manuscript presents a system (digital twin) for using data collected by citizen scientists to predict real time changes in species distributions. Using an app developed by the authors citizen scientists are able to submit their data directly to the system which then uses this data to update models the following day. The authors show that this method greatly improved predictions of species distributions especially for migrants. I really like this paper, the method and application are exciting and novel and very clearly explained throughout. The engagement by the public in this app is fantastic and as outlined it can greatly help educate people about the bird species in their environment while also collecting extremely useful data. By using several data collection methods they eliminate much of the bias associated with collecting such data and this seems like an important step forward. I can not find any flaws in this manuscript which would prevent its publication, and I find that the system/method and results presented would be very interesting to people in this field. Although I am not able to follow all the equations and algebra within the methods section, I find the logic presented behind how the model is structured and implemented sound. I do not have any suggestions for improvements for this work (which is very unlike me).

Catrina Morrison

We thank you very much for the (unusually) positive feedback.

Reviewer #1:

Remarks to the Author:

Thank you for responding to and addressing my comments. I have no further comments.

We thank the reviewer for the positive assessment.

Reviewer #2:

Remarks to the Author:

The author's have address most of my concerns for this manuscript. I have a few more:

-I would recommend removing the word "merely" in front of data integration, which strongly understates the methodological work in model-based data integration, which in many cases focuses on statistical inference and parameter interpretation. I think there is better language to distinguish data integration literature from DT methods.

We have removed "merely".

-"By relying solely on machine learning based bird classifications rather than citizen-based classifications, we remove an important part of observer heterogeneity and increase inclusivity by enabling ordinary citizens without bird identification skills to take part in data collection by making bird recordings."

I am not a fan of this sentence. Machine learning based bird classifications (generally speaking) are wrought with potential challenges (and biases) in practice. Recording applications such as Merlin filter classification to only "possible birds" based on historical rarity of occurrence. Machine learning based classifications are also prone to the same types of misclassification that occur by human ears (e.g., imitation calls of mockingbirds, blue jays imitating a red-shouldered hawk, etc.). Weather conditions make ML classification even more challenging, and experienced participants will often correctly identify birds that a recording using ML will not detect. Further, I am not convinced that machine learning based classification increases inclusivity. Recording applications (e.g., Merlin) have increased inclusivity encouraging more people to engage in bird watching, but I think it is odd to claim that machine learning classifications increases inclusivity. Further, (in general) for participants who use recording applications designed to help with identification, it is not necessarily clear whether or not recorded observer classifications relied on help from a ML-based recording application.

In this comment, I am not claiming that machine learning based classification as opposed to participant classification is worse or better. This statement understates the potential quality of participant based classification and how it can in some cases be better than and ML based classification. If I were the authors, I would downplay this choice as much as possible unless they have evidence that, in their application, ML based classification improved ___ performance in some way over participant based classification.

With this sentence, we are not comparing whether machine learning based classifications are better or worse than citizen science-based classifications; they both have their pros and cons as discussed in many parts of the manuscript. We consider the sentence accurate and non-opinionated for the following reasons. First, when we say “By relying solely on machine learning based bird classifications rather than citizen-based classifications, we remove an important part of observer heterogeneity”, we mean that when machine learning identifies birds, variation in the observers in their identification skills does not influence the data, and hence an important part of observer heterogeneity is removed. Second, when we continue by “...and increase inclusivity by enabling ordinary citizens without bird identification skills to take part in data collection by making bird recordings”, we mean that those citizens who cannot identify species at all by themselves can contribute equally valid data as those who can. This is because the citizens simply conduct recordings with the app rather than identify birds themselves. In our view this clearly increases inclusivity as anyone can contribute to data collection.